# R&B: Domain Regrouping and Data Mixture Balancing for Efficient Foundation Model Training

## Abstract

While data mixing strategies have successfully reduced training costs, existing methods suffer from two critical flaws: they rely on predetermined data domains that may fail to capture semantic nuances, and they scale computationally with the number of domains in a prohibitive way. We address these challenges by paying a fixed one-time cost to repartition source data into semantically similar domains and reusing training gradients to estimate domain importance. Thus, we propose R&B, a two-stage framework that re-partitions training data based on semantic similarity (Regroup) to create finer-grained domains, then efficiently optimizes the data composition (Balance) by leveraging a Gram matrix induced by domain gradients obtained throughout training. Unlike prior works, R&B removes the need for additional compute to obtain evaluation information such as losses or gradients. We analyze this technique under standard regularity conditions and provide theoretical insights that justify R&B's effectiveness compared to non-adaptive mixing approaches. Empirically, we demonstrate the effectiveness of R&B on five diverse datasets ranging from natural language to reasoning and multimodal tasks. With as little as 0.01% additional compute overhead, R&B matches or exceeds the performance of state-of-the-art data mixing strategies.

## 1 Introduction

The composition of training data directly impacts large language models' generalization abilities and robustness to diverse inputs. By changing the proportions of training data—a process known as a **data mixture optimization**—we can build models with comparable or superior performance with significantly fewer computational resources.

A wide variety of data mixture optimization techniques have been proposed (Fan et al., 2024b; Xie et al., 2023a; Chen et al., 2023; 2024; Jiang et al., 2024). These adjust the relative proportions ("mixture") of training data from different predefined domains, also known as *skills*. Skills are often assigned to data based on human judgments or on source metadata (Wettig et al., 2025). For example, in an instruction-tuning dataset, skills might include domain categories like open-question answering or summarization. For other datasets, skills may be defined based on where the data was scraped from (Wikipedia, StackOverflow, etc.). We find, however, that these coarse human-defined categorizations fail to capture the optimal groupings for data mixing. That is, ***human categorizations of skills are suboptimal when used for the development of LLM capabilities***.

Consider the Dolly-15k instruction-tuning dataset, which categorizes its data into general domains such as open-question answering, information extraction, and summarization (Conover et al., 2023). Rather than directly optimizing these coarse categories, our approach first re-partitions the data into finer-grained, semantically-grouped skills (Fig. 1, left). Optimizing the proportions across these semantically-clustered skills can significantly improve training performance over that of the general predefined domains. These improvements are even more pronounced when the number of skill partitions is optimized (Fig. 1, right). However, this more granular semantic-based clustering approach has a *critical drawback*. As the number of skills increases, prior data mixing methods become computationally prohibitive. Existing approaches typically require additional evaluations—either through forward passes over evaluation datasets for each skill or by computing per-skill gradient in-

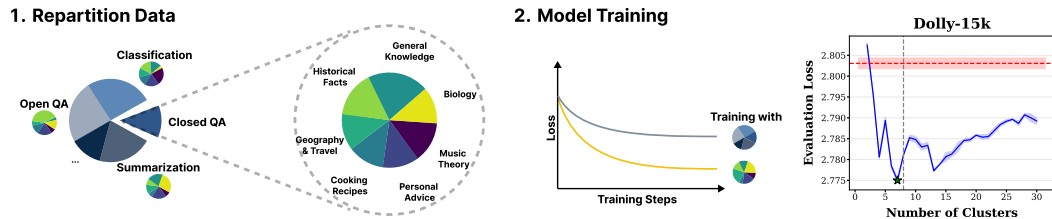

Figure 1: Instead of training with pre-determined domains (e.g., by task type) on the Dolly Instruction dataset, we find that it is often better to first repartition the data into finer-grained, semantically related domains. Optimizing both the proportions (middle) as well as the number of domains (right) of these new semantic domains can significantly improve model performance.

formation derived from target tasks. To overcome this limitation, **we propose an efficient gradient-based approach that leverages information already computed during training, bypassing the need for these expensive evaluations.**

These insights motivate our approach, **Regroup & Balance** (R&B), a two-stage framework for efficient data mixture optimization. First, we repartition (Regroup) training data into semantically coherent clusters based on embedding similarity. Then, we dynamically optimize domain weights (Balance) to capture individual domain contributions and cross-domain relationships, leveraging domain gradients computed throughout normal training. This produces the *best of all worlds*: it unlocks the performance gains in fine-grained skill clusters while dramatically reducing compute.

Theoretically, we derive insights to characterize the importance of semantic-based clustering in data mixing strategies and the optimality of the R&B method. Empirically, we validate R&B across five diverse data settings, encompassing natural-language, instruction-following, reasoning, and multi-modal tasks. *R&B only requires an additional 0.01% compute overhead, which cuts computational FLOPs by more than 99% relative to existing approaches, all while maintaining or improving model performance.*

We summarize our contributions as follows:

1. We establish that semantic-based categorizations of skills are superior to human-defined categories for foundation model data mixing algorithms (Section 3.1).
2. We introduce R&B, an efficient and theoretically justified two-stage framework that first repartitions data into semantically coherent clusters of skills, then dynamically reweights skill mixtures using their gradients (Sections 3, 3.1, and 3.3).
3. We theoretically and empirically demonstrate that R&B scales effectively with increased skill counts (Section 4).

## 2 RELATED WORK

We briefly discuss relevant related work:

**Group-level Data Mixing.** Prior work falls into two categories: static methods, which learn domain proportions before training, and dynamic methods, which adjust them as training progresses. Among the former, Fan et al. (2024b) uses a smaller model as a proxy to find mixture weights based on each domain's learning contribution, then applies them for the full model. Xie et al. (2023a) optimizes worst-case excess loss with a reference and proxy model, treating the resulting weights as domain proportions. These methods, though efficient, often neglect interactions between domains. Among the latter methods, Chen et al. (2023) model cross-domain interactions with a pre-built skills graph to optimize data composition. Building on it, Chen et al. (2024) introduces an online method that estimates domain interactions using training history information. However, as the domain count rises, these methods become computationally expensive.

**Sample-level Data Selection.** These techniques aim to improve dataset quality by evaluating individual samples. Several methods have been proposed, including using gradient alignment (Engstrom et al., 2024; Xia et al., 2024; Killamsetty et al., 2021; Huang et al., 2025), reward functions

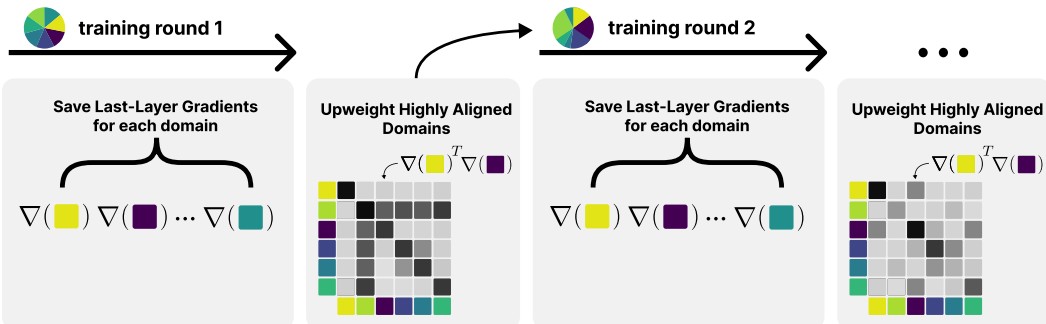

Figure 2: After repartitioning the dataset, we proceed with training as normal and store last-layer gradients obtained from each domain. After a fixed number of iterations, we compute a Gram matrix between all pairs of gradients, upweighting domains with high alignment to other domains.

(Wu et al., 2024), object detection models (Huang et al., 2024), and semantic similarity metrics (Xie et al., 2023b). A related line of work is through deduplication, which removes redundant or nearly identical examples using clustering (Abbas et al., 2023; Lee et al., 2022; Tirumala et al., 2023). Our work integrates these perspectives, repartitioning data into finer-grained groups while simultaneously optimizing domain mixtures.

**Scaling Laws.** These works model and predict performance when training on diverse domains. Ge et al. (2024) develop a bivariate scaling law to jointly model domain proportion and data volume, while Ye et al. (2024) propose a composite exponential law that accounts for cross-domain interactions. Liu et al. (2024) approach the problem empirically by training small models with varying mixtures to fit a predictive regression model, Kang et al. (2024) build on this to derive compute-optimal data mixtures. Roberts et al. (2025) points out that under a static data mix, knowledge and code skills have different compute-optima that can be aligned via data selection. R&B's dynamic allocation approach suggests the need for new scaling models that can capture the effects of adaptive data allocation strategies.

## 3 R&B: REGROUP AND BALANCE DATA MIXTURES

We first provide some context and intuition. We will refer to skills and domains interchangeably throughout this paper. In data mixing, each data source/skill/domain is assigned a proportion weight from the probability simplex, and data is sampled according to this probability distribution. Formally, for $m$ skills, we sample data according to the distribution $\boldsymbol{p} = [p_1, \ldots, p_m] \in \Delta^{m-1}$. We seek to answer two questions:

**R1. How should we define domains for data mixing?** Given a dataset, we wish to group the data into $m$ partitions suitable for data mixing strategies. We define a mapping function $S : \mathcal{X} \to \{1, 2, \ldots, m\}$ which labels each data point $x$ to its corresponding partition index. For a given dataset $D$, we have that $D_i = \{x \in D : S(x) = i\}$, and $\bigcup_{i=1}^{m} D_i = D$.

Intuitively, one would like to slice the data to minimize noise within each partition and reduce overlap between partitions. If perfect separation is possible, each partition would correspond to a distinct skill or capability domain, allowing for more targeted optimization of mixing weights. On the other hand, if each data sample is i.i.d. assigned to a group, *then we would not expect any data mixing strategy to be better than stratified sampling.*

**R2. How do we efficiently reweight domains?** Once $D_{\text{train}}$ has been partitioned into $m$ groups, our secondary objective is determining the optimal weight proportions $\boldsymbol{p}$ for each domain. Weights may change over time, so training is split into $T$ rounds. At the end of each round $t$, we can reweight the skills/domains $\boldsymbol{p}^{t+1} = [p_1^{t+1}, \ldots, p_m^{t+1}] \in \Delta^{m-1}$, and resume training according to the new proportions.

### 3.1 PROBLEM SETUP

We formulate our framework as a bilevel optimization problem for minimizing test loss on a dataset. The lower-level optimization aims to find the best training proportions $\boldsymbol{p}^t \in \Delta^{m-1}$ for each training

round $t \in 1, \ldots, T$. The upper-level problem seeks to find the best partitioning of a dataset $D$ into $m$ partitions $D_1, \ldots, D_m$ where $D = \bigcup_{i=1}^{m} D_m$. Let $D_{\text{eval},i} = \{x \in D_{\text{eval}} : S(x) = i\}$ be the set of evaluation points assigned to skill $i$ by $S$. Let $f_{\theta_t}$ be the model parametrized by $\theta_t$ that is trained during round $t$, i.e. $f_{\theta_{t+1}}$ is obtained by training $f_{\theta_t}$ with proportions $\boldsymbol{p}^t$. Formally, we aim to solve the following problem,

$$\min_{m \in \mathbb{Z}_+} \min_{\boldsymbol{p}^1, \ldots, \boldsymbol{p}^T \in \Delta^{m-1}} \mathcal{L}_{\text{eval}}(f_{\theta_{T+1}}), \tag{1}$$

where $\mathcal{L}_{\text{eval}}(f_{\theta_{T+1}})$ is the average evaluation loss for the partition $D_{\text{eval},i}$ after training model $f_{\theta_T}$ on mixture proportions $\boldsymbol{p}^T$ to obtain $f_{\theta_{T+1}}$.

Solving the full bilevel problem (1) is infeasible because for each candidate solution of the upper-level optimization problem, we must train a model for the lower-level optimization problem to obtain the loss. Thus, we propose decomposing Equation 1 into two:

$$S^*, m^* = \arg\min_{S,m} \mathcal{L}_{\text{clustering}}(S; f_{\text{Unif}}(D)), \tag{2}$$

$$\min_{\boldsymbol{p}^1, \ldots, \boldsymbol{p}^t \in \Delta^{m^*-1}} \mathcal{L}_{\text{eval}}^*(f_{\theta_{T+1}}). \tag{3}$$

In (2), we use $f_{\text{Unif}}$, which we denote as a model trained on fixed uniform proportions across skills to convergence (i.e. stratified sampling, $f_{\text{Unif}} = \arg\min_f \mathcal{L}_{\text{eval}}(f)$). This minimization is taken over a family of partitioning schemes $S$, such as the $S$ found through k-means, on the gradients of trained model $\mathcal{L}_{\text{eval}}(f_{\text{Unif}})$. In (3), we use the optimal choice of $m^*$ and partitioning scheme $S^*$ found in the previous stage, and solve the optimization problem at every training round $t$.

## 3.2 DEFINING DOMAINS

We investigate how to partition a given dataset to achieve optimal data mixing. Intuitively, data points that belong in a cluster should have a similar effect, i.e. gradients, during training. If gradients are not aligned, then swapping points with another cluster would reduce noise in both clusters. This leads to our first definition.

**Definition 1.** *A skill-assigning function $S : D \to [m]$ is stable in the direction $\nabla\mathcal{L}(\theta_t; \mathcal{D}_{\boldsymbol{p}})$ if for the skill $i = \arg\max_{i \in [m]} \nabla\mathcal{L}(\theta_t; D_i)^\top \nabla\mathcal{L}(\theta_t; \mathcal{D}_{\boldsymbol{p}})$ and any other $j \in [m]$, exchanging a pair $x_i \in D_i$, $x_j \in D_j$ does not improve $\nabla\mathcal{L}(\theta_t; D_i)^\top \nabla\mathcal{L}(\theta_t; \mathcal{D}_{\boldsymbol{p}})$.*

In other words, a clustering is said to be stable if no swapping of points improves alignment with the evaluation gradient. This definition provides a theoretical foundation for optimal data mixing, but it is still impractical to discover good groupings. The following lemma characterizes the maximum regret from swapping points between clusters.

**Lemma 3.1.** *Define the regret $R_S(i, j)$ under the skill-assigning function $S$ for class $j$ as the difference between the gradient alignments:*

$$R_S(i,j) = \max_{\tilde{D}_i \subset D_i \cup D_j, |\tilde{D}_i| = |D_i|} \nabla\mathcal{L}(\theta_t; \mathcal{D}_{\boldsymbol{p}})^\top \nabla\mathcal{L}(\theta_t, \tilde{D}_i) - \nabla\mathcal{L}(\theta_t; \mathcal{D}_{\boldsymbol{p}})^\top \nabla\mathcal{L}(\theta_t; D_i).$$

*Let $i, j \in |m|$ and assume $|D_i| = |D_j|$ and $\nabla\mathcal{L}(\theta_t; D_i)^\top \nabla\mathcal{L}(\theta_t; \mathcal{D}_{\boldsymbol{p}}) \geq \nabla\mathcal{L}(\theta_t; D_j)^\top \nabla\mathcal{L}(\theta_t; \mathcal{D}_{\boldsymbol{p}})$, and let $r_i = \max_{x \in D_i} |\nabla\mathcal{L}(\theta_t, x)^\top \nabla\mathcal{L}(\theta_t; \mathcal{D}_{\boldsymbol{p}}) - \nabla\mathcal{L}(\theta_t; D_i)^\top \nabla\mathcal{L}(\theta_t; \mathcal{D}_{\boldsymbol{p}})|$. Then we have*

$$R_S(i,j) \leq \max\left\{0, \frac{1}{2}\left(r_i + r_j - (\nabla\mathcal{L}(\theta_t; D_i)^\top \nabla\mathcal{L}(\theta_t; \mathcal{D}_{\boldsymbol{p}}) - \nabla\mathcal{L}(\theta_t; D_j)^\top \nabla\mathcal{L}(\theta_t; \mathcal{D}_{\boldsymbol{p}}))\right)\right\}.$$

While $R_S(i, j)$ is still dependent on the direction of the evaluation gradient $\nabla\mathcal{L}(\theta_t; \mathcal{D}_{\boldsymbol{p}})$, this result shows that for a clustering which assigns each class to have a small radius in every direction (an upper bound on $r_i$) and a large separation between their means, in many directions mostly orthogonal to their difference vector, $R_S(i, j)$ is 0. **This bound reveals additional structure that enables us to determine an effective clustering: clusters should have minimal radii while maintaining sufficient separation between their centroids**. This is equivalent to $S$ being stable, a property which R&B uses to achieve optimality as compared to using other clusterings (see Section 3.4 and note that $\nabla\mathcal{L}(\theta_t; D_i)^\top \nabla\mathcal{L}(\theta_t; \mathcal{D}_{\boldsymbol{p}})$ should be maximized).

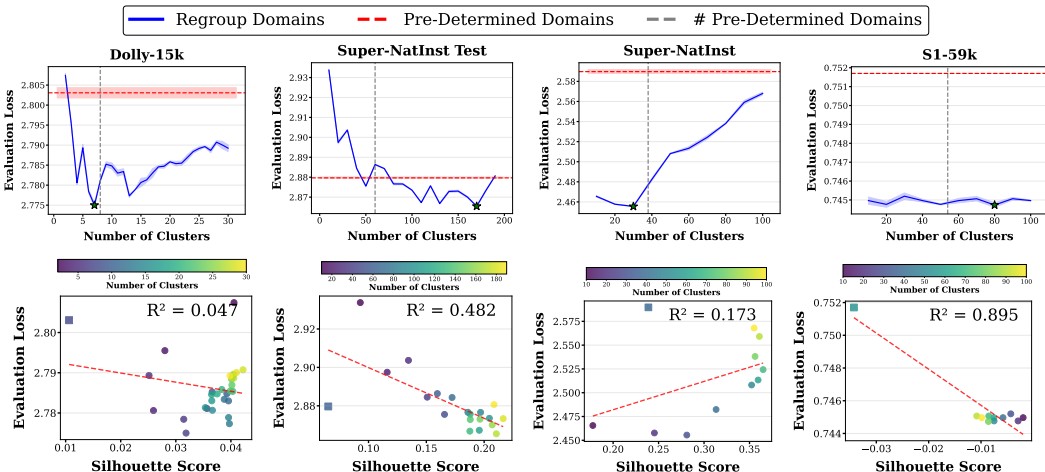

Figure 3: Across various data settings, we find that there is a "sweet spot" in the number of domains used for data mixing, indicated by the green star. The optimal number of groups varies significantly with the dataset, which motivates the need for compute-efficient data mixing. Additionally, we find that silhouette score often correlates with model performance, suggesting that it is possible to predict data mixing performance based on clustering metrics.

In light of these theoretical findings, we seek to empirically validate our claims by clustering real data and train using fixed proportions $p$. Our hypothesis is that well-clustered data can result in better overall training performance. In practice, there are many choices for the skill-assigning function $S$ and the number of skills. To keep our investigation tractable, we focus on k-means clustering, and sweep over $k$. We first embed our examples with ModernBERT-embed (Nussbaum et al., 2024), a state-of-the-art embedding model that supports long-context inputs. Then, we apply k-means and train a model using a uniform proportion of $k$ clusters. In Section 4, we evaluate our setup across four settings: Dolly-15k (Conover et al., 2023), Super Natural Instructions (Super-NatInst) & Super Natural Instruction Test (Super-NatInst Test) (Wang et al., 2022), and S1-Reasoning (Muennighoff et al., 2025), and across 3 seeds.

The top row of Figure 3 shows that **training on the resulting clusters often results in significantly better performance** compared to pre-determined partitions. On 3 of the 4 datasets, there is a U-shaped pattern in the number of clusters versus evaluation loss. Thus in many cases, there is an optimal choice of $k$—but it varies significantly between datasets.

Are these optimal clusters compact and well-separated, as our theory suggests? We find that generally, the answer is *yes*. We plot the silhouette score (Rousseeuw, 1987) of each cluster group against the final evaluation loss of the model. The bottom row of Figure 3 shows that on 3 of the 4 datasets, there is moderate to strong correlation between the clusters' silhouette score and model performance. **These results validate our theoretical insights that clusters which are well-separated result in better data mixing performance**. Furthermore, this suggests that it is possible to choose the optimal $k$ without training a model, which would lead to further cost savings.

### 3.3 PROPOSED METHOD

The R&B algorithm (Algorithm 1) performs adaptive data selection for efficient model training on partitioned datasets.

During each training round, R&B accumulates final-layer gradients from sampled batches and tracks which clusters contribute to model updates. Then it constructs a gradient similarity matrix that captures how gradients from different clusters relate to each other. This similarity information is combined with predefined evaluation proportions to produce an updated sampling distribution through a softmax operation. As training progresses, the algorithm adaptively shifts sampling probability toward clusters that contain the most valuable training examples, improving efficiency while main-

---

**Algorithm 1** R&B: Online Domain Data Selection

---

1: **Input:** Domain datasets $\{D^*_{\text{train},i}, D^*_{\text{eval},i}\}_{i=1}^{m^*}$, model $\theta \in \mathbb{R}^n$, training rounds $T$, steps per round $K$, evaluation weights $\boldsymbol{p} \in \mathbb{R}^{m^*}$
2: Initialize sampling distribution $\boldsymbol{p}^0 \leftarrow \text{Uniform}(m^*)$
3: **for** $t = 0$ to $T - 1$ **do**
4:     Initialize gradient accumulators $\nabla\mathcal{L}(\theta; D_i) \leftarrow \mathbf{0}_n$ and sample sets $\mathcal{S}_i \leftarrow \emptyset$ for all $i$
5:     **for** $k = 0$ to $K - 1$ **do**
6:         Sample mini-batch $\mathcal{B}$ from $D^*_{\text{train},i}$ using domain probabilities $\boldsymbol{p}^t$
7:         Update model: $\theta \leftarrow \theta - \eta\nabla\mathcal{L}(\theta; \mathcal{B})$
8:         **for** each domain $i$ present in $\mathcal{B}$ **do**
9:             Accumulate final-layer gradients: $\nabla\mathcal{L}(\theta; D_i)\ \texttt{+=}\ \nabla\mathcal{L}(\theta; \mathcal{B} \cap D^*_{\text{train},i})$
10:           Add samples to set: $\mathcal{S}_i \leftarrow \mathcal{S}_i \cup (\mathcal{B} \cap D^*_{\text{train},i})$
11:     Compute similarity matrix $G$ with $G_{ij} = \frac{1}{|\mathcal{S}_i||\mathcal{S}_j|}\nabla\mathcal{L}(\theta; D_i)^\top\nabla\mathcal{L}(\theta; D_j)$
12:     Update distribution: $\boldsymbol{p}^{t+1} \leftarrow \text{softmax}(\lambda G\boldsymbol{p}/\|G\boldsymbol{p}\|_2)$
13: **Return:** Final model parameters $\theta$

---

taining performance across all partitions. We provide a visual depiction of this procedure in Figure 2.

The key innovation of R&B lies in its use of gradient information to dynamically adjust sampling priorities, enabling models to learn effectively from heterogeneous data *without requiring extensive tuning for each data partition*.

## 3.4 Determining Optimal Proportions

Starting from the objective in (3), we greedily aim to find the best $\boldsymbol{p}^t$ such that $f_{\theta_{t+1}}$ has the greatest decrease in loss possible for this iteration of gradient descent. Specifically, $\theta_{t+1}$ is taken to be the SGD update of the current $\theta_t$, where $\theta_{t+1}(\boldsymbol{p}^t) = \theta_t - \eta\nabla\mathcal{L}(\theta_t; \mathcal{D}_{\boldsymbol{p}^t})$. Here, we have control over the weighting of the gradients $\boldsymbol{p}^t$ in this gradient descent step. This gradient term is the weighted sum of the gradients per skill: $\nabla\mathcal{L}(\theta_t; \mathcal{D}_{\boldsymbol{p}^t}) = \sum_i(\boldsymbol{p}^t)_i\nabla\mathcal{L}(\theta_t; \mathcal{D}_i)$. Now, assume that $\nabla\mathcal{L}$ is well approximated by its first order Taylor expansion (for some exposition on how this method behaves under a Lipschitz loss, see Appendix B.1). In this case, the optimization objective for finding the best mixture weights becomes $\boldsymbol{p}^t =$

$$\underset{\boldsymbol{p}' \in \Delta^{m-1}}{\arg\min}\ \mathcal{L}(\theta_t; \mathcal{D}_{\boldsymbol{p}}) - \nabla\mathcal{L}(\theta_t; \mathcal{D}_{\boldsymbol{p}})^\top\nabla\mathcal{L}(\theta_t; \mathcal{D}_{\boldsymbol{p}'}) = \underset{\boldsymbol{p}' \in \Delta^{m-1}}{\arg\max} \sum_{i=1}^{m}(\boldsymbol{p}')_i\nabla\mathcal{L}(\theta_t; \mathcal{D}_{\boldsymbol{p}})^\top\nabla\mathcal{L}(\theta_t; \mathcal{D}_i).$$

This objective is simply a linear objective over the simplex, which will be maximized by one of the corners of the simplex; this maximal corner will correspond with the skill that aligns most with the target loss averaged over skills with distribution $\boldsymbol{p}$.

This optimum, however, will be highly discontinuous and will only ever be able to sample from a single skill per descent step. This creates two major issues: a highly variable sampling scheme may cause highly variable and unpredictable behavior in training, and the per-gradient sampling scheme required for our method requires samples from each skill within the training batch. We address this by adding cross entropy regularization, which prevents extreme mixing proportions and aligns better with scaling law results (Ye et al., 2024). The regularized solution remains tractable:

$$\boldsymbol{p}^t = \frac{1}{Z}\text{softmax}\left(\frac{\lambda}{\|G\boldsymbol{p}\|_2}G\boldsymbol{p}\right),$$

where $Z$ is a normalization constant, $\lambda$ is a hyperparameter, and $G_{ij} = \nabla\mathcal{L}(\theta_t; \mathcal{D}_i)^\top\nabla\mathcal{L}(\theta_t; \mathcal{D}_j)$. See Appendix B.1 for derivation details.

**Comparison to Multiplicative Weights and DoGE.** Other works (Fan et al. (2024b), Chen et al. (2023)) use an update rule based on multiplicative weights. The update in DoGE (Fan et al., 2024b) (which is most similar to ours) is written in our notation as:

$$\boldsymbol{p}'_t = 1/Z(\boldsymbol{p}'_{t-1}\exp(\eta W_t/\mu)) = 1/Z(\boldsymbol{p}'_{i-1}\exp(\eta G_t\boldsymbol{p}/\mu)) = 1/Z\left(\exp\left(\eta\sum_{i=1}^{t}G_i\boldsymbol{p}/\mu\right)\right),$$

where $\mu = 1/\lambda$ and $W_t = G\boldsymbol{p}$ by the definitions of $G$ and $\boldsymbol{p}$. Note that $G_i\boldsymbol{p}$ is aggregated up to round $t$. In contrast, our method aggregates $G_i\boldsymbol{p}$ for $K$ training iterations before updating $\boldsymbol{p}'$, and then overwrites the mixture weights to be those of the previous window. This prevents the diminishing influence of later updates and gradient scale reduction that occurs in multiplicative weights approaches. An update that refreshes the proportions at every window will circumvent the strong bias towards the optimal weights remaining similar, as was also used in Chen et al. (2023).

## 4 EXPERIMENTS

We study the effectiveness of R&B empirically across a diverse range of datasets and tasks. Our experiments aim to validate the following claims:

**C1.** R&B can *match or improve training performance* on natural language tasks while *significantly reducing computational overhead* compared to existing methods.
**C2.** R&B can improve training performance *beyond natural language modalities*.

### 4.1 DATA MIXING ON NATURAL LANGUAGE TASKS

Table 1: Compute overhead incurred from regrouping (left) and data mixing (right). Across three datasets, R&B **significantly reduces the compute overhead** on additional evaluation compared to existing methods, while **matching or exceeding performance.**

| Method | % Overhead (OH) |
|---|---|
| Sup-NatInst | 21.3 |
| Sup-NatInst test | 18.9 |
| Dolly | 3.4 |
| S1K-1.1 | 0.1 |

| Method | SUP-NATINST $(m = 38)$ | | SUP-NATINST test $(m = 60)$ | | Dolly-15k $(m = 8)$ | |
|---|---|---|---|---|---|---|
| | Loss | % OH ($\downarrow$) | Loss | % OH ($\downarrow$) | Loss | % OH ($\downarrow$) |
| Stratified | 2.591 | 0 | 2.877 | 0 | 2.788 | 0 |
| Skill-It | 2.632 | 595.5 | 2.911 | $6 \times 10^7$ | 2.786 | 14.46 |
| Aioli | 2.622 | 1336.5 | 2.883 | $7 \times 10^6$ | 2.779 | 62.5 |
| DGA | 2.591 | 1.723 | 2.893 | 1601 | 2.787 | 0.41 |
| R&B | $(m^* = 30)$ | | $(m^* = 100)$ | | $(m^* = 7)$ | |
| | **2.381** | **0.009** | **2.859** | **0.1** | **2.765** | **0.0006** |

**Setup.** We evaluate R&B against four established baselines: stratified sampling, Skill-It (Chen et al., 2023), Aioli (Chen et al., 2024), and DGA (Fan et al., 2024a). The comparison involves three natural-language datasets: Dolly-15k (Conover et al., 2023), NaturalInstructions In-domain SUP-NATINST (Wang et al., 2022), and NaturalInstructions Test SUP-NATINST. For fine-tuning and evaluation, we employ GPT-Neo 125M models (Black et al., 2021); full experimental details are provided in Appendix F. We measure the final evaluation loss to assess framework effectiveness and calculate the relative compute overhead (relative to standard training) incurred from re-estimating proportions. We separate the compute cost of repartitioning (Table 1 left) from domain reweighting (Table 1 right) to provide fair comparison between different data mixing methods. The formulas for the relative compute overhead are derived in Appendix D.

**Results.** Table 1 right demonstrates R&B's superior performance across all datasets. On SUP-NATINST In-domain, R&B achieves the lowest evaluation loss of 2.381 with minimal computational overhead (0.009%). On SUP-NATINST Test, R&B outperforms all methods with a loss of 2.859 and an overhead of just 0.1%, compared to Skill-It ($6 \times 10^7$%) and Aioli ($7 \times 10^6$%). For Dolly-15k, R&B delivers competitive performance (loss: 2.765) against Aioli (2.779), with significantly lower overhead (0.0006% versus 62.5%). As expected from C1, **R&B consistently delivers strong results with orders of magnitude better computational efficiency** than other data mixing approaches.

**Ablations.** Table 2 examines the effect of semantic regrouping across data mixing strategies. On SUP-NATINST In-domain, **regrouping improves performance for most methods** (with gains of 5.3 to 8.1%) except Skill-It. Moreover, this **justifies the one-time compute cost** of repartitioning the data, because it is compatible with multiple data mixing methods to provide better performance. On SUP-NATINST Test, regrouping provides modest benefits for all methods except Aioli, which experiences a slight performance decline. For Dolly-15k, regrouping consistently improves performance

Table 2: Regrouping skills before applying data mixing strategies can yield substantial improvements. Underlined values indicate where regrouping beats the original grouping for that method and dataset. **Highlighted** values (with brown background) indicate the best overall performance for each dataset. Note that we do not apply Balance to the original categorization of SUP-NATINST Test, as we assume that training data and validation data are bucketed into the same $m$ groups.

| **Method** | SUP-NATINST | | SUP-NATINST Test | | Dolly-15k | |
|---|---|---|---|---|---|---|
| | Original | Regroup | Original | Regroup | Original | Regroup |
| $m$ | 38 | 30 | 60 | 100 | 8 | 7 |
| Stratified | 2.591 | 2.454 | 2.877 | 2.871 | 2.788 | 2.761 |
| Skill-It | 2.632 | 2.812 | 2.911 | 2.881 | 2.786 | 2.778 |
| Aioli | 2.622 | 2.488 | 2.883 | 2.947 | 2.779 | **2.760** |
| DGA | 2.591 | 2.453 | 2.893 | 2.871 | 2.787 | 2.761 |
| Balance | 2.520 | **2.381** | - | **2.859** | 2.783 | 2.765 |

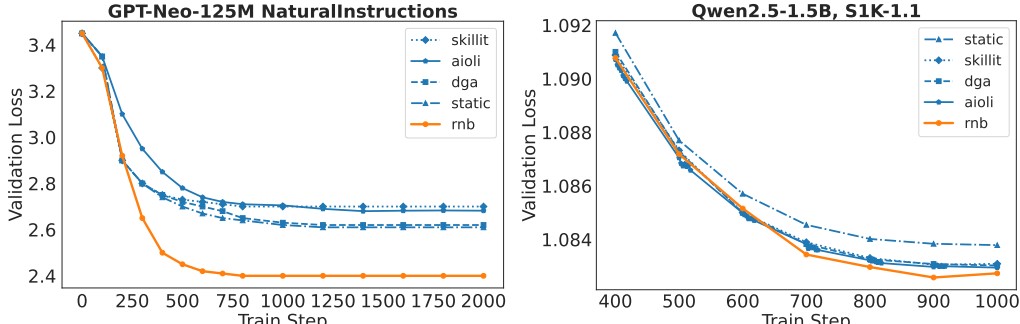

Figure 4: Evaluation loss over training steps. **Left:** GPT-Neo 125M on NaturalInstructions. **Right:** Qwen2.5-1.5B on S1k-1.1. Both settings demonstrates the effectiveness of R&B and its impact on convergence efficiency.

across all methods. While regrouping is often beneficial, it is not universally effective, as seen in Skill-It's performance drop on SUP-NATINST and Aioli's minor regression on SUP-NATINST Test. Notably, the Balance method combined with regrouping, referred to as R&B, delivers the best overall performance on both SUP-NATINST datasets, while Aioli with the help of regrouping achieves the highest performance on Dolly-15k.

Even without regrouping, **our gradient-based Balance method demonstrates strong performance while maintaining minimal overhead**. On the original SUP-NATINST In-domain dataset, Balance method helps achieving a evaluation loss of 2.520, significantly outperforming other data-mixing methods. On Dolly-15k, it reaches a competitive loss of 2.783. We omit Balance results for original SUP-NATINST Test since our method requires training and validation data to share the same $m$ groups—a limitation that can be easily addressed by re-mapping validation points to corresponding training skills.

Finally, Figure 4 demonstrates R&B's convergence efficiency. R&B reaches convergence with only 20% of the training steps required by other methods, while also achieving lower final loss values.

## 4.2 BEYOND LANGUAGE MODELING

Next, we examine tasks beyond the scope of our approach's primary design, including language reasoning and multimodal applications.

**Reasoning Setup.** We investigate whether optimized data mixtures can enhance model performance on reasoning tasks. We use the S1 dataset (Muennighoff et al., 2025), containing about 1.1k reasoning traces from challenging math problems, drawn from 54 distinct sources. The s1 dataset is

Table 3: R&B performs better over stratified sampling method in more complex modalities.

| $m$ | Method | ImageNet | ImageNet dist. shift | VTAB | Retrieval | Avg over 38 datasets ($\uparrow$) |
|---|---|---|---|---|---|---|
| 10 | Stratified | 0.034 | 0.044 | 0.157 | 0.104 | **0.146** |
| | R&B | 0.033 | 0.040 | 0.153 | 0.104 | 0.141 |
| 20 | Stratified | 0.036 | 0.044 | 0.153 | 0.106 | 0.145 |
| | R&B | 0.031 | 0.042 | 0.163 | 0.103 | **0.148** |
| 50 | Stratified | 0.042 | 0.047 | 0.170 | 0.107 | 0.153 |
| | R&B | 0.042 | 0.047 | 0.177 | 0.108 | **0.158** |
| 100 | Stratified | 0.034 | 0.043 | 0.152 | 0.107 | 0.139 |
| | R&B | 0.041 | 0.047 | 0.151 | 0.104 | **0.145** |
| 150 | Stratified | 0.034 | 0.043 | 0.165 | 0.109 | 0.143 |
| | R&B | 0.039 | 0.050 | 0.164 | 0.109 | **0.153** |

divided into 10 groups. For fine-tuning, we employ the Qwen2.5-1.5B model (Yang et al., 2024) as a representative example.

**Reasoning Results.** Figure 4 shows that R&B yields faster convergence and achieves lower evaluation loss than other data mixing methods. Notably, R&B requires only 70% of the training steps needed by the stratified method. This supports our claim C2, confirming that R&B is a versatile method applicable beyond language modeling.

**Multimodal Setup.** We extend our setup to include multimodal tasks. We train CLIP models (Radford et al., 2021) from scratch using the small-scale DataComp dataset (Ilharco et al., 2021; Gadre et al., 2023). Our dataset comprises approximately 10 million image-caption pairs sourced from the web.[1] To ensure dataset quality, we select the top 30% of samples based on CLIP Score (Gadre et al., 2023), retaining 3.8 million high-quality pairs. We extract image embeddings for both the filtered training dataset and DataComp's evaluation benchmark, which spans 38 diverse downstream tasks. We apply k-means clustering to repartition data into varying numbers of groups. We use DataComp's training configurations and adopt R&B as our training method. And, we use stratified sampling as our baseline method.

**Results.** We presented CLIP models' performance in Table 3. R&B outperforms stratified sampling when the number of domains exceeds 10. With 50 domains, R&B achieves a 3.27% relative improvement over the stratified sampling baseline. These findings highlight the effectiveness of R&B when the number of underlying domains is potentially high and validate its extensibility to modalities beyond natural language, confirming C2.

## 5 CONCLUSION

In this paper, we introduced Regroup & Balance (R&B), a two-stage framework that breaks free from two fundamental constraints found in state-of-the-art data mixing strategies: the limitations of predetermined domains and the computational bottleneck of per-skill evaluations. Empirically, R&B matched or exceeded state-of-the-art data mixing methods while requiring two orders of magnitude less compute overhead. By reimagining both domain groups and their mixtures, R&B charts a more efficient path forward for foundation model training in an era of unlimited unstructured data and constrained computational resources.

---

[1]Some URLs provided by DataComp are now broken. See here for details.

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

The appendix is structured as follows. Appendix A introduces our notation, followed by theoretical insights and proofs in Appendix B. Algorithmic details are provided in Appendix C. We then analyze the computational cost of existing data mixing methods in Appendix D. Implementation specifics and experimental setups are detailed in Appendix E and Appendix F, respectively. Appendix G presents the results of our ablation studies. Lastly, we give interpretation for determined domains in Appendix H.

## A    NOTATION

| Symbol | Meaning |
|--------|---------|
| $d$ | The dimensionality of each skill (toy theory) |
| $m$ | The number of skills |
| $[\cdot]$ | The sequence of numbers $1, 2, \ldots, n$ |
| $\Delta^{m-1}$ | The $m-1$ dimensional simplex |
| $\mathcal{D}$ | A mixture data distribution |
| $\mathcal{D}_i$ | The distribution for skill $i$ |
| $\mathcal{D}_{\boldsymbol{p}}$ | The mixture of $\mathcal{D}_i$ according to $\boldsymbol{p}$ |
| $D_i$ | A sample from the distribution $\mathcal{D}_i$ |
| $G$ | The inner product between gradients of different skills $\nabla\mathcal{L}(\theta; \mathcal{D}_i)$ |
| $\mathcal{L}(\theta; \mathcal{D})$ | The expected loss of $\theta$ over $\mathcal{D}_{\boldsymbol{p}}$ |
| $\nabla\mathcal{L}(\theta; \mathcal{D}_i)$ | The skill gradient for skill $i$ |
| $\boldsymbol{p}$ | The evaluation data proportions |
| $\boldsymbol{p}'$ | The chosen training data proportions |
| $S$ | A skill-assigning function $S : \mathcal{X} \to [m]$ |

## B    THEORETICAL RESULTS

The goal is to find the best data mixture throughout training. With the many degrees of freedom such an algorithm can take, a few assumptions are made. First, an update is restricted to being performed by SGD with the following update rule:

$$\theta_{t+1} = \theta_t - \eta\nabla\mathcal{L}(\theta_t; \mathcal{D}_{\boldsymbol{p}'}).$$

The modification from standard SGD is the ability to change $\boldsymbol{p}'$ which allows for a different sampling mixture.

### B.1    METHOD DERIVATION

The design of the proportion finding algorithm described here comes from one core assumption: the gradient update works roughly linearly. Specifically, we assume, for a small enough ball around $\theta_t$,

$$\mathcal{L}(\theta; \mathcal{D}_{\boldsymbol{p}}) \approx \mathcal{L}(\theta_t; \mathcal{D}_{\boldsymbol{p}}) + \nabla\mathcal{L}(\theta_t; \mathcal{D}_{\boldsymbol{p}})^\top(\theta - \theta_t).$$

Inputting the SGD update rule,

$$\mathcal{L}(\theta_{t+1}; \mathcal{D}_{\boldsymbol{p}}) \approx \mathcal{L}(\theta_t; \mathcal{D}_{\boldsymbol{p}}) - \eta\nabla\mathcal{L}(\theta_t; \mathcal{D}_{\boldsymbol{p}})^\top\nabla\mathcal{L}(\theta_t; \mathcal{D}_{\boldsymbol{p}'}).$$

Since the gradient is linear, we can treat $\nabla\mathcal{L}(\theta_t; \mathcal{D}_{\boldsymbol{p}})$ as a weighted sum of $\nabla\mathcal{L}(\theta_t; \mathcal{D}_i)$, i.e. the individual skill gradients, based on the proportions $\boldsymbol{p}_i$. Define the Gram matrix $G$ as

$$G_{ij}^{(t)} = \nabla\mathcal{L}(\theta_t; \mathcal{D}_i)^\top\nabla\mathcal{L}(\theta_t; \mathcal{D}_j).$$

Note that $G^{(t)}$ is dependent on the iteration, but this will often be written only as $G$ for notational clarity. This matrix also has a clear interpretation as a neural tangent kernel (NTK) aggregated over each skill rather than over each data point as is commonly used. With this matrix and treating $\boldsymbol{p}$ and $\boldsymbol{p}'$ as vectors, our assumption simplifies to

$$\mathcal{L}(\theta_{t+1}; \mathcal{D}_{\boldsymbol{p}}) \approx \mathcal{L}(\theta_t; \mathcal{D}_{\boldsymbol{p}}) - \eta\boldsymbol{p}^\top G\boldsymbol{p}'.$$

Since the aim is to decrease loss as much as possible, we want to maximize $\eta\boldsymbol{p}^\top G\boldsymbol{p}'$ over $\boldsymbol{p}' \in \Delta^{m-1}$. This objective is linear over the simplex, so optima will only be found at the corners of

the simplex. However, this would imply only one class is sampled from. This causes a few issues. First, the per-skill gradient computation method requires samples from skill $i$ to find the gradient for skill $i$. No information about other skills will be gathered through training. Second, the gradient as a function of time will be discontinuous. As the maximum skill of the optimization changes, the gradient will instantly change to the skill gradient for the new maximal skill. We impose a common regularity to restrict $\boldsymbol{p}'$ to the simplex by solving for the following:

$$\boldsymbol{p}' = \arg\max_{\boldsymbol{p}' \in \Delta^{m-1}} \eta \boldsymbol{p}^\top G \boldsymbol{p}' - \lambda_1 \sum_{i=1}^m \boldsymbol{p}'_i \log \boldsymbol{p}'_i + \lambda_2 \sum_{i=1}^m \boldsymbol{p}'_i.$$

Equivalently, we can multiply the $\lambda_1$ and $\lambda_2$ terms by a constant without affecting the optimum. This constant $\|\eta G p\|_2$ is chosen to make the parameter $\lambda_1$ have a roughly equal effect between the cross entropy term of the base objective, regardless of $\eta$, $G$, and $\boldsymbol{p}$. This also means that as the model trains and the gradients decrease in magnitude, the effects of the regularization term will decrease. Otherwise, proportions will tend towards uniform. Now, solve for:

$$\boldsymbol{p}' = \arg\max_{\boldsymbol{p}' \in \Delta^{m-1}} \eta \boldsymbol{p}^\top G \boldsymbol{p}' - \lambda_1 \|\eta G \boldsymbol{p}\| \sum_{i=1}^m \boldsymbol{p}'_i \log \boldsymbol{p}'_i + \lambda_2 \|\eta G \boldsymbol{p}\| \sum_{i=1}^m \boldsymbol{p}'_i.$$

The parameter $\lambda_1$ acts as a normal hyper-parameter, but $\lambda_2$ is a Lagrange multiplier enforcing $\boldsymbol{p}'$ lay on the simplex. Taking a gradient and setting to 0,

$$0 = \eta G \boldsymbol{p} - \lambda_1 \|\eta G \boldsymbol{p}\| \log \boldsymbol{p}' + (\lambda_1 + \lambda_2)\|\eta G \boldsymbol{p}\| \mathbf{1}_m,$$

$$\log \boldsymbol{p}' = \frac{\eta}{\lambda_1 \|\eta G \boldsymbol{p}\|} G \boldsymbol{p} + \frac{\lambda_1 + \lambda_2}{\lambda_1} \mathbf{1}_m,$$

$$\boldsymbol{p}' = \exp\left(\frac{\eta}{\lambda_1 \|\eta G \boldsymbol{p}\|} G \boldsymbol{p}\right) \exp\left(\frac{\lambda_1 + \lambda_2}{\lambda_1}\right)$$

$$= \exp\left(\frac{1}{\lambda_1 \|G \boldsymbol{p}\|} G \boldsymbol{p}\right) \exp\left(\frac{\lambda_1 + \lambda_2}{\lambda_1}\right).$$

Here, $\mathbf{1}_m$ is the vector of all 1s of dimension $m$. Also, $\lambda_2$ will take on a value to make $\boldsymbol{p}'$ sum to 1. Thus, setting $Z = \sum_{i=1}^m \exp\left(\frac{1}{\lambda_1 \|G \boldsymbol{p}\|} G \boldsymbol{p}\right)_i$ and letting $\lambda = 1/\lambda_1$, we have

$$\boldsymbol{p}' = \frac{1}{Z} \exp\left(\frac{\lambda}{\|G \boldsymbol{p}\|} G \boldsymbol{p}\right)$$

$$= \text{softmax}(\frac{\lambda}{\|G \boldsymbol{p}\|} G \boldsymbol{p}).$$

Note that in this parametrization, a large $\lambda$ indicates a small penalty from the entropy term. This solution aligns with the unconstrained solution described above from finding the maximal corner of the simplex, except now the solution is smooth.

### B.1.1 MAX CAN BE BETTER THAN STATIC PROPORTIONS

This method, for a small enough learning rate, will result in a reduction of a smooth loss greater than any other fixed proportions.

**Lemma B.1.** *Let $\ell$ be an $L$-smooth loss function and $f_\theta$ is learned with SGD on datasets $\mathcal{D}_i$ with associated sampling prior $\boldsymbol{p}_i$, and let $\boldsymbol{p}''$ be some other fixed distribution of datasets. If the learning rate $\eta$ satisfies $\eta \leq \frac{1}{L}\left(\frac{\max_i \nabla\mathcal{L}(\theta_0;\mathcal{D}_i)^\top \nabla\mathcal{L}(\theta_0;\mathcal{D}_{\boldsymbol{p}}) - (\nabla\mathcal{L}(\theta_0;\mathcal{D}_{\boldsymbol{p}''}))^\top \nabla\mathcal{L}(\theta_0;\mathcal{D}_{\boldsymbol{p}})}{\max_i \|\nabla\mathcal{L}(\theta_0;\mathcal{D}_i)\|^2 + \|\nabla\mathcal{L}(\theta_0;\mathcal{D}_{\boldsymbol{p}''})\|^2}\right)$, then a gradient descent step with $\boldsymbol{p}' = \delta_{\arg\max_i(G\boldsymbol{p})_i}$ results in a greater (or equal) decrease in the loss than $\boldsymbol{p}''$.*

*Proof.* Let $\theta_0$ be some base parameter, $\boldsymbol{p}' = \delta_i$ for $i = \arg\max_i(G\boldsymbol{p})_i$, and let

$$\theta_{\boldsymbol{p}'} = \theta_0 - \eta \nabla\mathcal{L}(\theta_0; \mathcal{D}_{\boldsymbol{p}'}),$$
$$\theta_{\boldsymbol{p}''} = \theta_0 - \eta \nabla\mathcal{L}(\theta_0; \mathcal{D}_{\boldsymbol{p}''}).$$

Assume that $\mathcal{L}$ is $L$-smooth. Now consider

$$\mathcal{L}(\theta_{\boldsymbol{p}''};\mathcal{D}_{\boldsymbol{p}}) - \mathcal{L}(\theta_{\boldsymbol{p}'};\mathcal{D}_{\boldsymbol{p}}) = (\mathcal{L}(\theta_{\boldsymbol{p}''};\mathcal{D}_{\boldsymbol{p}}) - \mathcal{L}(\theta_0;\mathcal{D}_{\boldsymbol{p}})) - (\mathcal{L}(\theta_{\boldsymbol{p}'};\mathcal{D}_{\boldsymbol{p}}) - \mathcal{L}(\theta_0;\mathcal{D}_{\boldsymbol{p}}))$$

$$\geq -\eta \boldsymbol{p}^\top G \boldsymbol{p}'' + \eta \boldsymbol{p}^\top G \boldsymbol{p}' - \eta^2 L {\boldsymbol{p}'}^\top G \boldsymbol{p}' - \eta^2 L {\boldsymbol{p}''}^\top G \boldsymbol{p}''$$

$$= \eta(\max_i (G\boldsymbol{p})_i - \boldsymbol{p}^\top G \boldsymbol{p}'' - \eta L({\boldsymbol{p}'}^\top G \boldsymbol{p}' + {\boldsymbol{p}''}^\top G \boldsymbol{p}'')).$$

Now let $i = \arg\max_i (G\boldsymbol{p})_i$ and let $\eta \leq \frac{1}{L}\left(\frac{(G\boldsymbol{p})_i - \boldsymbol{p}^\top G\boldsymbol{p}''}{G_{ii} + {\boldsymbol{p}''}^\top G\boldsymbol{p}''}\right)$. This quantity is positive (and therefore well defined for $\eta \geq 0$ since $\boldsymbol{p}^\top G\boldsymbol{p}''$ is the convex combination of values all less than (or equal to) $(G\boldsymbol{p})_i$. Thus,

$$\mathcal{L}(\theta_{\boldsymbol{p}''};\mathcal{D}_{\boldsymbol{p}}) - \mathcal{L}(\theta_{\boldsymbol{p}'};\mathcal{D}_{\boldsymbol{p}}) \geq \eta((G\boldsymbol{p})_i - \boldsymbol{p}^\top G\boldsymbol{p}'' - \eta L(G_{ii} + {\boldsymbol{p}''}^\top G\boldsymbol{p}''))$$

$$\geq \eta((G\boldsymbol{p})_i - \boldsymbol{p}^\top G\boldsymbol{p}'' - \frac{1}{L}\left(\frac{(G\boldsymbol{p})_i - \boldsymbol{p}^\top G\boldsymbol{p}''}{G_{ii} + {\boldsymbol{p}''}^\top G\boldsymbol{p}''}\right) L(G_{ii} + {\boldsymbol{p}''}^\top G\boldsymbol{p}''))$$

$$= \eta((G\boldsymbol{p})_i - \boldsymbol{p}^\top G\boldsymbol{p}'' - ((G\boldsymbol{p})_i - \boldsymbol{p}^\top G\boldsymbol{p}''))$$

$$= 0.$$

Therefore, for a small enough learning rate, the choosing the gradient with the largest skill results in a larger decrease in the loss than using priors proportional to the evaluation data. The learning rate can also be bounded using gradient notation and taking a maximum over $G_{ii}$:

$$\eta \leq \frac{1}{L}\left(\frac{\max_i \nabla\mathcal{L}(\theta_{\theta_0};\mathcal{D}_i)^\top \nabla\mathcal{L}(\theta_{\theta_0};\mathcal{D}_{\boldsymbol{p}}) - (\nabla\mathcal{L}(\theta_{\theta_0};\mathcal{D}_{\boldsymbol{p}''}))^\top \nabla\mathcal{L}(\theta_{\theta_0};\mathcal{D}_{\boldsymbol{p}})}{\max_i \|\nabla\mathcal{L}(\theta_{\theta_0};\mathcal{D}_i)\|^2 + \|\nabla\mathcal{L}(\theta_0;\mathcal{D}_{\boldsymbol{p}''})\|^2}\right).$$

$\square$

### B.1.2 CLUSTERING

One major phenomenon observed here is that clustering the data points works well for domain mixing, and sometimes even outperforms the provided labels for the different skills. These clusters are taken via the embeddings of some other model, which are assumed to mimic the gradients of the model being learned.

When $\eta \approx 0$, the change in the loss can be well-approximated by its first order Taylor expansion:

$$\mathcal{L}(\theta_t;\mathcal{D}_{\boldsymbol{p}}) - \mathcal{L}(\theta_{t+1};\mathcal{D}_{\boldsymbol{p}}) \approx \eta \nabla\mathcal{L}(\theta_t;\mathcal{D}_{\boldsymbol{p}})^\top \nabla\mathcal{L}(\theta_t;\mathcal{D}_{\boldsymbol{p}'}).$$

For the following sections, let $D_i = \{x \in D : S(x) = i\}$ for some skill-assigning function $S$.

**Definition 2.** *A skill-assigning function $S : D \to [m]$ is stable in the direction $\nabla\mathcal{L}(\theta_t;\mathcal{D}_{\boldsymbol{p}})$ if for the skill $i = \arg\max_{i\in[m]} \nabla\mathcal{L}(\theta_t;D_i)^\top \nabla\mathcal{L}(\theta_t;\mathcal{D}_{\boldsymbol{p}})$ and any other $j \in [m]$, exchanging a pair $x_i \in D_i, x_j \in D_j$ does not improve $\nabla\mathcal{L}(\theta_t;D_i)^\top \nabla\mathcal{L}(\theta_t;\mathcal{D}_{\boldsymbol{p}})$.*

Intuitively, a stable clustering is one which doesn't improve $\nabla\mathcal{L}(\theta_t;D_i)^\top \nabla\mathcal{L}(\theta_t;\mathcal{D}_{\boldsymbol{p}})$ by exchanging points between classes.

**Lemma B.2.** *Let $S$ be some $\{0,1\}$ skill-assigning function on $D$ with $\nabla\mathcal{L}(\theta_t;D_0)^\top \nabla\mathcal{L}(\theta_t;\mathcal{D}_{\boldsymbol{p}}) \geq \nabla\mathcal{L}(\theta_t;D_1)^\top \nabla\mathcal{L}(\theta_t;\mathcal{D}_{\boldsymbol{p}})$ and let $x_0, x_1 \in D$ with $S(x_0) = 0$ and $S(x_1) = 1$, and let $\tilde{C}$ be the clustering identical to $S$ except on $x_0, x_1$ where each is assigned to the opposite class. Then, $\nabla\mathcal{L}(\theta_t;D_{\tilde{S},\boldsymbol{p}'})^\top \nabla\mathcal{L}(\theta_t;\mathcal{D}_{\boldsymbol{p}})^\top > \nabla\mathcal{L}(\theta_t;D_{S,\boldsymbol{p}'})^\top \nabla\mathcal{L}(\theta_t;\mathcal{D}_{\boldsymbol{p}})$ if $\nabla\mathcal{L}(\theta_t, x_1)^\top \nabla\mathcal{L}(\theta_t;\mathcal{D}_{\boldsymbol{p}}) > \nabla\mathcal{L}(\theta_t, x_0)^\top \nabla\mathcal{L}(\theta_t;\mathcal{D}_{\boldsymbol{p}})$.*

*Proof.* Let $D_i = \{x \in D : S(x) = i\}$, $\tilde{D}_i = \{x \in D : \tilde{S}(x) = i\}$, $n_i = |D_i|$, and $\boldsymbol{p}'$ be the proportions that put mass only on the maximal value of $A\nabla\mathcal{L}(\theta_t;\mathcal{D}_{\boldsymbol{p}})$. If we define

$$A = \begin{bmatrix} \nabla\mathcal{L}(\theta_t;D_0) \\ \nabla\mathcal{L}(\theta_t;D_1) \end{bmatrix},$$

then

$$\nabla\mathcal{L}(\theta_t; D_S)^\top \nabla\mathcal{L}(\theta_t; \mathcal{D}_{\boldsymbol{p}}) = \boldsymbol{p}'^\top A \nabla\mathcal{L}(\theta_t; \mathcal{D}_{\boldsymbol{p}}),$$

$$\nabla\mathcal{L}(\theta_t; D_{\tilde{S}})^\top \nabla\mathcal{L}(\theta_t; \mathcal{D}_{\boldsymbol{p}}) = \boldsymbol{p}'^\top \tilde{A} \nabla\mathcal{L}(\theta_t; \mathcal{D}_{\boldsymbol{p}})$$

$$= \boldsymbol{p}'^\top A \nabla\mathcal{L}(\theta_t; \mathcal{D}_{\boldsymbol{p}}) + \boldsymbol{p}'^\top \left[ \begin{matrix} \frac{1}{n_1}\nabla\mathcal{L}(\theta_t, x_1)^\top \nabla\mathcal{L}(\theta_t; \mathcal{D}_{\boldsymbol{p}}) - \frac{1}{n_1}\nabla\mathcal{L}(\theta_t, x_0)^\top \nabla\mathcal{L}(\theta_t; \mathcal{D}_{\boldsymbol{p}}) \\ \frac{1}{n_0}\nabla\mathcal{L}(\theta_t, x_0)^\top \nabla\mathcal{L}(\theta_t; \mathcal{D}_{\boldsymbol{p}}) - \frac{1}{n_0}\nabla\mathcal{L}(\theta_t, x_1)^\top \nabla\mathcal{L}(\theta_t; \mathcal{D}_{\boldsymbol{p}}) \end{matrix} \right].$$

Therefore swapping the classes of $x_0$ and $x_1$ results in an improvement for $\tilde{S}$ over $S$ if $\nabla\mathcal{L}(\theta_t, x_1)^\top \nabla\mathcal{L}(\theta_t; \mathcal{D}_{\boldsymbol{p}}) > \nabla\mathcal{L}(\theta_t, x_0)^\top \nabla\mathcal{L}(\theta_t; \mathcal{D}_{\boldsymbol{p}})$. $\qquad\square$

We immediately have the following corollary:

**Lemma B.3.** *A skill-assigning function* $S : D \to [m]$ *is stable in the direction* $\nabla\mathcal{L}(\theta_t; \mathcal{D}_{\boldsymbol{p}})$ *if for the skill* $i = \arg\max_{i \in [m]} \nabla\mathcal{L}(\theta_t; D_i)^\top \nabla\mathcal{L}(\theta_t; \mathcal{D}_{\boldsymbol{p}})$ *and any other* $j \in [m]$, *for all* $x_i \in D_i$, $x_j \in D_j$,

$$\nabla\mathcal{L}(\theta_t; \mathcal{D}_{\boldsymbol{p}})^\top \nabla\mathcal{L}(\theta_t, x_i) \geq \nabla\mathcal{L}(\theta_t; \mathcal{D}_{\boldsymbol{p}})^\top \nabla\mathcal{L}(\theta_t, x_j).$$

An important fact to note is that the evaluation gradient $\nabla\mathcal{L}(\theta_t; \mathcal{D}_{\boldsymbol{p}})$ can be arbitrary, especially if the evaluation and training data come from different distributions. To reduce the benefit of swapping points between classes, a good clustering will be stable in as many directions as possible.

A simple but noisy choice is to take all points in the convex hull to be in unique clusters, and all interior points to make up another cluster. While this clustering is stable in every direction, the classes are very small and therefore likely to be noisy, especially as training progresses and the gradient landscape shifts. A better alternative is clustering points if they can be linearly separated from the others.

Assume $D$ can be partitioned into $D_0$ and $D_1$ such that $f(x) = \text{sign}(\boldsymbol{v}^\top x + b)$ is a perfect classifier, so in the direction $\boldsymbol{v}$, $S$ which labels the data based on the partition is stable. If $f$ has a large margin, then many other $\boldsymbol{v}$ also a linear separators, and therefore also have $S$ stable.

This still may be too restrictive though in general settings where data points are more spread apart. Instead, it may be good to compare the regret of skill-labeling with a sub-optimal labeling.

**Definition 3.** *The regret* $R_S(i, j)$ *under the skill-assigning function* $S$ *for class* $j$ *is the difference between* $\max_{\tilde{D}_i \subset D_i \cup D_j, |\tilde{D}_i| = |D_i|} \nabla\mathcal{L}(\theta_t; \mathcal{D}_{\boldsymbol{p}})^\top \nabla\mathcal{L}(\theta_t; \tilde{D}_i)$ *and* $\nabla\mathcal{L}(\theta_t; \mathcal{D}_{\boldsymbol{p}})^\top \nabla\mathcal{L}(\theta_t; D_i)$.

This regret is exactly difference between the first-order loss decrease using $\tilde{D}_i$ as compared to $D_i$, where new elements in $\tilde{D}_i$ come from $D_j$.

**Lemma B.4.** *Let* $i, j \in |m|$ *and assume* $|D_i| = |D_j|$. *Assume* $\nabla\mathcal{L}(\theta_t; D_i)^\top \nabla\mathcal{L}(\theta_t; \mathcal{D}_{\boldsymbol{p}}) \geq \nabla\mathcal{L}(\theta_t; D_j)^\top \nabla\mathcal{L}(\theta_t; \mathcal{D}_{\boldsymbol{p}})$, *and let* $r_i = \max_{x \in D_i} |\nabla\mathcal{L}(\theta_t, x)^\top \nabla\mathcal{L}(\theta_t; \mathcal{D}_{\boldsymbol{p}}) - \nabla\mathcal{L}(\theta_t; D_i)^\top \nabla\mathcal{L}(\theta_t; \mathcal{D}_{\boldsymbol{p}})|$ *and similarly define* $r_j$. *Then*

$$R_S(i, j) \leq \max\{0, \frac{1}{2}(r_i + r_j - (\nabla\mathcal{L}(\theta_t; D_i)^\top \nabla\mathcal{L}(\theta_t; \mathcal{D}_{\boldsymbol{p}}) - \nabla\mathcal{L}(\theta_t; D_j)^\top \nabla\mathcal{L}(\theta_t; \mathcal{D}_{\boldsymbol{p}})))\}.$$

*Proof.* Let $R_i = \{\nabla\mathcal{L}(\theta_t, x)^\top \nabla\mathcal{L}(\theta_t; \mathcal{D}_{\boldsymbol{p}}) | x \in D_i\}$ and $R_j = \{\nabla\mathcal{L}(\theta_t, x)^\top \nabla\mathcal{L}(\theta_t; \mathcal{D}_{\boldsymbol{p}}) | x \in D_j\}$. Also, in this notation, $r_i = \max_{x \in R_i} |x - \mathbb{E}_{x \sim R_i}[x]|$, and define $\delta = \mathbb{E}_{x \in R_i}[x] - \mathbb{E}_{x \in R_j}[x]$. The problem then reduces to

$$R_S(i, j) \leq \max\{0, \frac{1}{2}(r_i + r_j - \delta)\}.$$

Also, $R_S(i, j)$ in this one dimensional case becomes the largest over $m \in [|R_i|]$ of the difference between the $m$ largest values of $R_j$ and the $m$ smallest values of $R_i$. This is maximized over all possible $R_i$ and $R_j$ when half of $R_i$ is $\mathbb{E}_{x \in R_i}[x] - r_i$ and the other half is $\mathbb{E}_{x \in R_i}[x] + r_i$, and similarly for $R_j$. The difference between the max $R_j$ and the min $R_i$ is $r_i + r_j - \delta$, and only half of these data points attain these max and min values, so $R_S(i, j) \leq \max\{0, \frac{1}{2}(r_i + r_j - \delta)\}$ as desired. $\qquad\square$

This extends the case where skills $i$ and $j$ are linearly separable in the direction $\nabla\mathcal{L}(\theta_t; \mathcal{D}_{\boldsymbol{p}})$. It further provides insight in how to pick skills. To reduce any pairwise regret, the radii $r_i$ and $r_j$ of the clusters from their mean should be as small as possible in every direction.

### B.1.3 OOD EVALUATION CLUSTERS

When performing k-means clustering, there is a choice in clustering the training points and then assigning the evaluation points, or clustering the evaluation points and then assigning the training points. The latter choice has a major problem: evaluation clusters may have no training points near them. This causes a major dilemma for the training procedure attempting to sample from a distribution that lacks any data points.

We adopt the former method of clustering based on the training points to circumvent this issue. However, a new issues arises: evaluation data may be OOD and have no representatives in the training data. The result is the label provided to those OOD points is the same as the closest training points. These can be quite distant and therefore not a strong representation of their gradient. However, this still is the optimal choice, as all other training points are a greater distance away and therefore have a weaker similarity. This label assignment also adds more weight to the class that is most aligned with the OOD evaluation data, increasing its sample rate to learn both the ID and OOD data for that class.

## C  ALGORITHM DETAILS

We fully outline our algorithms for solving the optimization problems specified in Equation 2 and Equation 3, respectively. And we provide our Algorithm details in Algorithm 2 and Algorithm 3.

---

**Algorithm 2** R&B Skill Partitioning

1: **Input:** Training data $D_{\text{train}}$, Evaluation data $D_{\text{eval}}$, Embedding model $\psi : \mathcal{X} \to \mathbb{R}^d$,
2: Clustering algorithm $cluster : \mathcal{P}(\mathbb{R}^d) \times \mathcal{K} \to \mathbb{N} \times (\mathbb{R}^d \to \mathbb{N})$,
3: Clustering metric $metric : (\mathbb{R}^d \to \mathbb{N}) \times \mathcal{P}(\mathbb{R}^d) \to \mathbb{R}$,
4: Range of clustering hyperparameters $K \in \mathcal{K}$
5: **Output:** Optimal number of clusters $m^*$, Optimal mapping function $f^*$, Partitioned datasets
   $\{D^*_{\text{train},i}\}_{i=1}^{m^*}, \{D^*_{\text{eval},i}\}_{i=1}^{m^*}$
6: $D_{\text{train}} \leftarrow \{\psi(x) : x \in D_{\text{train}}\}$  ▷ Collect embeddings for training data
7: $D_{\text{eval}} \leftarrow \{\psi(x) : x \in D_{\text{eval}}\}$  ▷ Collect embeddings for eval data
8: **for** $k \in K$ **do**
9:     $m, f = cluster(D_{\text{train}}, k)$
10:     $score = metric(f, D_{\text{train}})$
11:     $m^*, f^*, score^* = \arg\max_{m,f,score}(score, score^*)$
12: **for** $i = 1$ to $m^*$ **do**
13:     $D^*_{\text{train},i} \leftarrow \{x \in D_{\text{train}} : f^*(x) = i\}$  ▷ Partition training data
14:     $D^*_{\text{eval},i} \leftarrow \{y \in D_{\text{eval}} : f^*(y) = i\}$  ▷ Partition evaluation data
15: **Return** $m^*, f^*, \{D^*_{\text{train},i}\}_{i=1}^{m^*}, \{D^*_{\text{eval},i}\}_{i=1}^{m^*}$

---

## D  COMPUTE COST MODELS FOR ONLINE DATA MIXING

We formalize a cost model for estimating the amount of compute required for several data mixing methods. Table 4 reports cost in terms of FLOPs, or number of floating point operations required to perform each method.

Following Kaplan et al. (2020), we will use the estimate for the compute cost $C = C_{\text{forward}} + C_{\text{backward}} \approx 2ND + 4ND$, where $N$ is the number of model parameters, and $D$ is the number of training tokens. Here, we also make use of the empirical observation that the amount of compute for a backward pass is roughly twice that of the amount for a forward pass (Hobbhahn, 2021).

For all methods analyzed, we make the following assumptions:

- Each method trains on $D_t$ tokens across $m$ domains,
- Each method has access to an evaluation dataset with $D_e$ tokens,
- Training is divided into $T$ rounds with domain reweighting between rounds,

---

**Algorithm 3** R&B Online Data Selection

---

1: **Input:** Partitioned datasets $\{D^*_{\text{train},i}\}_{i=1}^{m^*}$, $\{D^*_{\text{eval},i}\}_{i=1}^{m^*}$, model parameters $\theta \in \mathbb{R}^n$, training rounds $T$, steps per round $K$, evaluation proportions $\boldsymbol{p}' \in \mathbb{R}^{m^*}$
2: Initialize sampling distribution $\boldsymbol{p}^0 = \text{Uniform}(m^*)$
3: **for** $t = 0, \ldots, T-1$ **do**
4:      **for** $i \in [m^*]$ **do**
5:          $\nabla \mathcal{L}(\theta; D_i) \leftarrow \boldsymbol{0}_n$               ▷ Initialize gradient accumulator for domain $i$
6:          $\mathcal{S}_i \leftarrow \emptyset$                      ▷ Initialize set of samples from domain $i$
7:      **for** $k = 0, \ldots, K-1$ **do**
8:          Sample batch $\mathcal{D} = \{x_j\}_{j=1}^B$ where $x_j \sim D^*_{\text{train},i}$ with $i \sim \boldsymbol{p}^t$ for each $j$
9:          $\theta_{tK+k+1} \leftarrow \theta_{tK+k} - \eta \nabla \mathcal{L}(\theta_{tK+k}; \mathcal{D})$
10:         **for** $i \in \{f^*(d) : d \in \mathcal{D}\}$ **do**
11:            $\nabla \mathcal{L}(\theta; D_i) \leftarrow \nabla \mathcal{L}(\theta; D_i) + \nabla \mathcal{L}(\theta_{tK+k}; \mathcal{D} \cap D^*_{\text{train},i})$
12:            $\mathcal{S}_i \leftarrow \mathcal{S}_i \cup (\mathcal{D} \cap D^*_{\text{train},i})$
13:      Construct $G \in \mathbb{R}^{m^* \times m^*}$ where $G_{ij} = \frac{1}{|\mathcal{S}_i||\mathcal{S}_j|} \nabla \mathcal{L}(\theta; D_i)^T \nabla \mathcal{L}(\theta; D_j)$
14:      $\boldsymbol{p}^{t+1} \leftarrow \text{softmax}(\eta \lambda G \boldsymbol{p}')$
15: **Return**

---

Table 4: Computational cost comparison of data mixing methods. We report (1) total cost of training, given under the table Total Compute Cost, and relative compute overhead over standard training. Standard training requires no additional compute overhead since its proportions are fixed. In the common setting where the number of skills $m$ is much smaller than that of evaluation tokens $D_e$ and training tokens $D_t$, R&B enjoys superior computational efficiency.

| Method | Total Compute Cost (FLOPs) | Relative Compute Overhead (vs. Standard Training) |
|---|---|---|
| Standard Training | $6D_t N$ | $0$ |
| Skill-It (Chen et al., 2023) | $6(1 + m\delta)D_t N + 2(T + m)D_e N$ | $m\delta + \frac{(T+m)D_e}{3D_t}$ |
| Aioli (Chen et al., 2024) | $6D_t N + 2(Tm)D_e N$ | $\frac{TmD_e}{3D_t}$ |
| DGA (Fan et al., 2024a) | $6(1 + m\delta)D_t N + 6T(\delta D_e)N$ | $m\delta + T\delta \frac{D_e}{D_t}$ |
| R&B (Ours) | $6D_t N + Tm^2 N$ | $\frac{Tm^2}{6D_t}$ |

- Each method uses some fraction of the training dataset, $\delta \cdot D_t$ (where $\delta < 1$), to perform their reweighting procedure.

For our analysis, it is necessary to split the forward and backward compute costs because the data mixing algorithms we study involve a domain-reweighting mechanism that requires model evaluation on a hold-out dataset. Model evaluation only requires a forward pass, whereas model training requires both a forward and backward pass. To illustrate this point, let $D_{train}$ be the number of tokens in the training dataset, while $D_{eval}$ is the number of tokens in the evaluation dataset. Training a model on all available training data has a total cost of $6ND_{train}$, while computing model evaluation once has a cost of $2ND_{eval}$.

### D.1 CLUSTERING

In Table 1 we computed the compute cost overhead compared to standard training. To compute the relative cost, we have the formula

$$\frac{2N_{embed}D_{train,embed}}{6N_{model}D_{train}}$$

where $N_{embed}$ is the size of the embedding model, $D_{train,embed}$ is the number of tokens used for embedding each example, and $D_{train}$ is the number of tokens seen during training. Since we are

using ModernBERT-base, we have $N_{embed} = 149,014,272$. For NaturalInstructions and Dolly datasets, $N_{model} = 125,000,000$, and for S1, $N_{model} = 1,500,000,000$.

When computing $D_{train,embed}$, we find that embedding only the tokens of the question (rather than the full response) is sufficient for creating well-separated clusters. For $D_{train}$, we calculate by multiplying the number of training steps by batch size by context length for each example (concrete numbers are given in Table 5).

For $D_{train,embed}$ and $D_{train}$, we have calculated the number of tokens in each dataset as follows:

| Dataset | $D_{train,embed}$ | $D_{train}$ |
|---|---|---|
| Sup Nat-Inst | 8,710,588 | 16,384,000 |
| Sup Nat-Inst Test | 7,707,944 | 16,384,000 |
| Dolly-15k | 1,957,372 | 16,384,000 |
| S1 | 534,639 | 16,384,000 |

### D.2 SKILL-IT

Skill-It (Chen et al., 2023) has two stages in its data-mixing procedure: estimating a graph $A$ which is used as part of its domain reweighting procedure, and training itself.

For learning $A$, a model is trained on each of $m$ domains for some fraction of $D_{train}$, then evaluated on $D_{eval}$. For comparative purposes, we will assume that $\delta D_{train}$ training tokens are used in this process of constructing $A$, for $\delta < 1$. Furthermore, we assume these tokens are divided evenly among each of $m$ domains. Then the compute cost for learning $A$ is $6(\delta D_{train})N + 2(mD_{eval})N$. Training is split into $T$ rounds, and after each round, the model is re-evaluated on $D_{eval}$ to update the domain proportions. The compute cost for training, then, is $6(D_{train})N + 2TD_{eval}N$. This brings the total compute cost to

$$6(1+\delta)D_{train}N + 2(T+m)D_{eval}N.$$

### D.3 AIOLI

Similar to Skill-It, Aioli (Chen et al., 2024) also includes two stages for learning $A$ and training, but incorporates both directly into the training process. At a high level, training is also split into $T$ rounds, where each round dedicates some fraction $\delta < 1$ to learning $A$. When learning $A$, the model is trained on each of $m$ domains sequentially, and re-evaluating the resulting model on the evaluation dataset. Consequently, the training compute cost for learning $A$ is simply absorbed into the overall cost for training, but the model still must be evaluated on $D_{eval}$ for each domain. Within a round, this process repeats for the number of sweeps $k$, but here we will set $k = 1$ to simplify the analysis. Therefore, the total compute cost for training improves to $6(D_{train})N$, but the compute cost for evaluation increases to $2(Tm)D_{eval}N$. This brings the total compute cost to

$$6D_{train}N + 2(Tm)D_{eval}N.$$

### D.4 DGA

Dynamic Gradient Alignment (Fan et al., 2024a) instead uses gradient information to reweight the domain proportions. Their method splits training into $T$ rounds, and reweights proportions after each round. Their procedure involves sampling a batch from each domain, and then performing a forward and backward pass to obtain gradients respective of each domain. They then obtain gradients for a batch on a specific dataset $D_{spe}$ (which for consistency of analysis we will simply refer to as $D_{eval}$), and computes the inner product between the gradients of each domain and that of $D_{eval}$. In order to equalize model performance with Skill-It and Aioli, we will assume that a batch from each training domain contains $\frac{\delta}{m}D_{train}$ tokens, and a batch from the specific dataset contains $D_{eval}$ tokens. Then, computing each domain's gradient has a cost of $6(\frac{\delta}{m})D_{train}N$, and computing the specific dataset's gradient has a cost of $6D_{eval}N$. We assume that computing the inner product between two model gradients is linear in $N$ so there is an additional $mN$ compute overhead. Therefore, the total compute cost is

$$6(1+\delta)D_{train}N + 6T(D_{eval} + m)N.$$

## D.5 R&B (OURS)

Similar to all above methods, we split training into $T$ rounds, and reweight domain proportions at the end of each round. Like Dynamic Gradient Alignment, we opt to use gradient inner product information to inform our reweighting procedure. Crucially, however, we make two observations: (1) gradients per domain can be collected on the fly during normal backpropagation, and (2) our optimization problem only requires knowledge about the respective proportions of $D_{eval}$, and does not use gradient or loss information about $D_{eval}$. Instead, we simply compute the equivalent of matrix $A$ which is a Gram matrix comprised of the inner products between the gradients of each respective domain. As a result, the compute cost of training our method is simply $6(D_{train})N$, and the compute cost of evaluation is just $m^2N$. Therefore, the total compute cost is

$$6(D_{train})N + m^2N.$$

**Efficiency Analysis.** Under typical conditions where the number of skills is much smaller than the size of the evaluation dataset, **R&B demonstrates superior computational efficiency**. Its evaluation overhead scales only with $m^2$ rather than with $D_e$, making it particularly advantageous for scenarios with large evaluation datasets but a moderate number of domains.

When comparing specifically with DGA, R&B's advantage depends on the relationship between the number of domains and evaluation data size. R&B is more efficient when $m < \sqrt{D_e}$, which holds in most practical settings. Even when $m$ approaches or exceeds $\sqrt{D_e}$, R&B maintains partial efficiency benefits through its 6× lower coefficient on the evaluation term, and by avoiding the additional $\delta$ fraction for computing gradients.

## E  IMPLEMENTATION DETAILS

We start with an explanation of gradient computations.

### E.1  EFFICIENT GRADIENT COMPUTATION

Standard training pipelines provide per-batch gradients, but we need per-example gradients in order to aggregate per-skill gradients for our method. We perform a gradient decomposition similar to the method introduced in Goodfellow (2015) to efficiently circumvent this. A simple application of the chain-rule means we can exactly recover per-example gradients of a linear layer in a mini-batch with just one backwards pass by multiplying an example's input with that mini-batch's gradient.

Adopting the notational convention from Wang et al. (2024), let $\mathbf{s} = \mathbf{a}\mathbf{W}$ be a linear layer where $\mathbf{W} \in \mathbb{R}^{d_1 \times d_2}$ is a weight matrix, $\mathbf{a} = (\mathbf{a}^{(1)}, \dots, \mathbf{a}^{(B)})^\top \in \mathbb{R}^{B \times d_1}$ is the input to the mini-batch, and $\mathbf{s} = (\mathbf{s}^{(1)}, \dots, \mathbf{s}^{(B)})^\top \in \mathbb{R}^{B \times d_2}$ is the layer's pre-activation output. Denote by $\ell^{(i)}$ the loss on the $i^{\text{th}}$ example in the mini-batch. Let $\ell$ denote the summed loss of the mini-batch. It follows from the chain rule that the gradient of $\ell^{(i)}$ with respect to $\mathbf{W}$ can be expressed as

$$\frac{\partial \ell^{(i)}}{\partial \mathbf{W}} = \frac{\partial \ell^{(i)}}{\partial s^{(i)}} \frac{\partial s^{(i)}}{\partial \mathbf{W}} = \frac{\partial \ell^{(i)}}{\partial s^{(i)}} a^{(i)} = \frac{\partial \ell}{\partial s^{(i)}} a^{(i)},$$

where the last equality follows from the fact that the $\frac{\partial \ell^{(j)}}{\partial s^{(i)}}$ terms disappear when $i \neq j$. Notably, the $\frac{\partial \ell}{\partial s^{(i)}}$ term is available through standard training, and $a^{(i)}$ can be easily tracked. We aggregate per-example gradients into their respective skills, allowing for efficient per-skill gradient computation.

## F  EXPERIMENTAL DETAILS

We evaluate our method, R&B, against four baseline data mixing methods: Stratified sampling, Skill-It, Aioli, and DGA (Dynamic Gradient Alignment). We conducted experiments on three datasets of varying sizes and characteristics.

### F.1  DATASETS

- DOLLY-15K: An instruction follow-up dataset consisting of 15,000 examples with eight original skill categories.

- SUP-NATINST (Natural Instructions In-Domain): A 285k dataset created from Natural Instructions by selecting 100 tasks out of 876 available tasks containing 38 original skill categories.
- SUP-NATINST-Test (Natural Instructions Out-of-Domain): A 3.56M dataset created from Natural Instructions with questions and answers from domains not seen SUP-NATINST, containing 60 original skill categories.

For in-distribution datasets (NI-ID and Dolly-15k), we use 90% of the total dataset for training and 5% for testing. For regrouping experiments, we generate embeddings using ModernBERT with a dimension of 786 and cluster the datasets using k-means.

## F.2 EXPERIMENTAL CONFIGURATIONS

Table 5: Experimental Settings Across Different Datasets

| Parameter | Dolly-15k, NI-ID, NI-OOD | S1-1.1k |
|---|---|---|
| Model | GPT-Neo 125M | Qwen2-1.5B |
| Training batch size | 16 | 2 |
| Evaluation batch size | 16 | 2 |
| Context Length | 512 | 8192 |
| Learning rate | 5e-5 | 1e-5 |
| Training Steps | 2000 | 1000 |
| Optimizer | AdamW | AdamW |

## G  EXTENDED TRAINING RESULTS

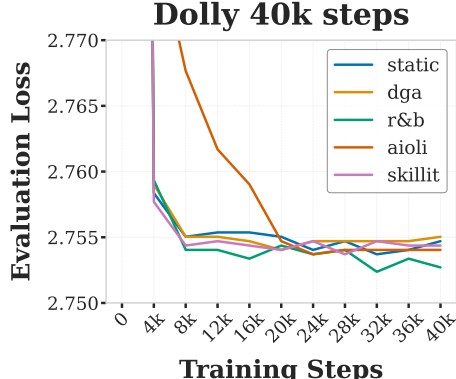

| Method | Evaluation Loss |
|---|---|
| Stratified | 2.733 |
| DGA | 2.733 |
| Aioli | 2.724 |
| Skill-It | 2.728 |
| R&B (ours) | **2.723** |

Figure 5: Left: Training loss curves for Dolly-15k trained for 40,000 steps with different data mixing methods using the original category partitioning. Right: Average test loss on Dolly-15k after 40,000 training steps using original category partitioning. **Highlighted** values (with brown background) indicate the best overall performance.

In this appendix, we provide additional experimental results for training on the Dolly-15k dataset for an extended period of 40,000 steps. This allows us to understand the long-term behavior of R&B compared to different data mixing methods. Figure 5 shows the training loss curves for different data mixing methods over the full 40,000 steps (left) alongside the final evaluation loss after 40,000 steps (right). R&B maintains consistent performance over other data mixing methods, demonstrating the stability of our approach and achieving the best performance with a loss of 2.723.

For this extended training experiment, we focus on the original category partitioning (rather than our regrouping approach) to demonstrate R&B's effectiveness even with pre-defined categories when

Table 6: Training budget allocations and experimental settings for all datasets and methods.

| Dataset | Domains | Method | Training Steps | Method-Specific Settings |
|---------|---------|--------|----------------|--------------------------|
| Dolly-15k | Original (8) | Stratified | 2000 | full eval dataset |
| | | Skill-It | 2000 | 200 steps for graph estimation, full eval dataset |
| | | Aioli | 2000 | rounds=2, sweeps=1, full eval dataset |
| | | DGA | 2000 | full eval dataset |
| | | R&B | 2000 | full eval dataset |
| | Regrouped (7) | Stratified | 2000 | full eval dataset |
| | | Skill-It | 2000 | 200 steps for graph estimation, full eval dataset |
| | | Aioli | 2000 | rounds=2, sweeps=1, full eval dataset |
| | | DGA | 2000 | full eval dataset |
| | | R&B | 2000 | full eval dataset |
| NI-ID | Original (38) | Stratified | 2000 | full eval dataset |
| | | Skill-It | 2000 | 200 steps for graph estimation, full eval dataset |
| | | Aioli | 2000 | rounds=2, sweeps=1, full eval dataset |
| | | DGA | 2000 | full eval dataset |
| | | R&B | 2000 | full eval dataset |
| | Regrouped (30) | Stratified | 2000 | full eval dataset |
| | | Skill-It | 2000 | 200 steps for graph estimation, full eval dataset |
| | | Aioli | 2000 | rounds=2, sweeps=1, full eval dataset |
| | | DGA | 2000 | full eval dataset |
| | | R&B | 2000 | full eval dataset |
| NI-OOD | Original (60) | Stratified | 2000 | full eval dataset |
| | | Skill-It | 2000 | 200 steps for graph, full eval dataset |
| | | Aioli | 2000 | rounds=1, sweeps=1, 50k eval samples |
| | | DGA | 2000 | full eval dataset |
| | | R&B | - | - |
| | Regrouped (100) | Stratified | 2000 | full eval dataset |
| | | Skill-It | 1000 | 25 steps for graph, 10k eval samples |
| | | Aioli | 1000 | rounds=1, sweeps=1, 50k eval samples |
| | | DGA | 2000 | full eval dataset |
| | | R&B | 2000 | full eval dataset |

\* Note: There is no result for R&B in the NI-OOD original column because the method requires finding training skills in the evaluation dataset. In out-of-domain settings, test skills and train skills are different, causing the Gp_norm in R&B to be NaN.

Table 7: Training budget allocations and experimental settings for S1-59K dataset.

| Dataset | Domains | Method | Training Steps | Method-Specific Settings |
|---------|---------|--------|----------------|--------------------------|
| S1-59K | Original (6) | Stratified | 500 | full eval dataset |
| | | R&B | 500 | num_layers_to_track=1, lamda=3, full eval dataset |
| | Regrouped (10) | Stratified | 500 | full eval dataset |
| | | R&B | 500 | num_layers_to_track=1, lamda=3, full eval dataset |

given sufficient training time. We observe that all data mixing methods eventually converge to similar performance levels after sufficient training, but R&B maintains a consistent advantage throughout the training process. This suggests that our gradient-based approach effectively captures the optimal training dynamics from early stages, leading to more efficient parameter updates throughout the training process.

Furthermore, we study how R&B reweights proportions over time. As illustrated in Figure 6, our method employs a dynamic approach to domain importance throughout the training process. Initially, domain weights fluctuate significantly as the model explores the contribution of each domain to overall performance. By the midpoint of training (around steps 40-60), a clear pattern emerges with Domains 1 and 5 receiving substantially higher weights (reaching approximately 0.225) compared to other domains. Notably, while these weights gradually trend toward the evaluation distribution proportions shown in Figure 6, they never completely converge to match the actual evaluation proportions. For instance, Domain 1 maintains a training weight of around 0.225 even though its

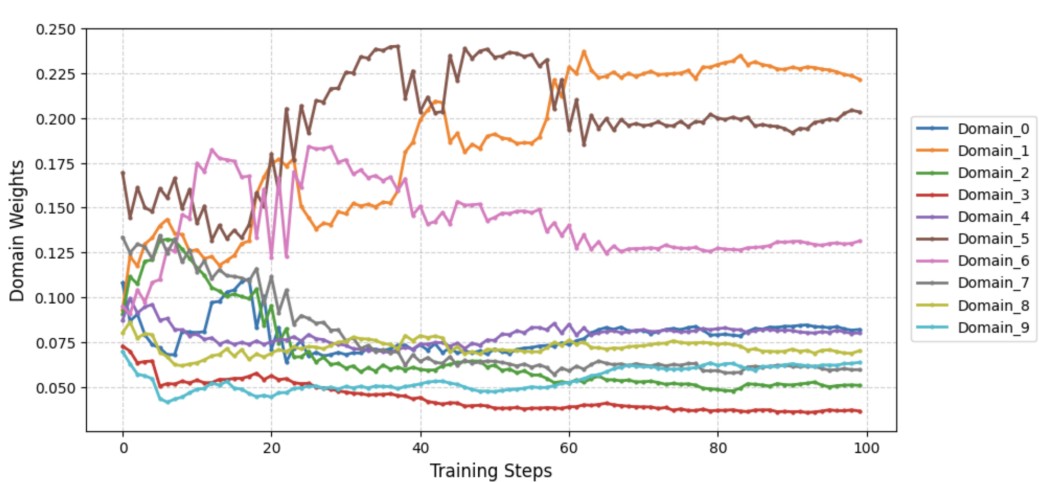

Figure 6: Domain weight evolution during training. Our method dynamically adjusts the importance of each domain throughout the training process, with Domains 1 and 5 eventually receiving the highest weights while Domains 0, 2, 3, 7, 8, and 9 are downweighted over time.

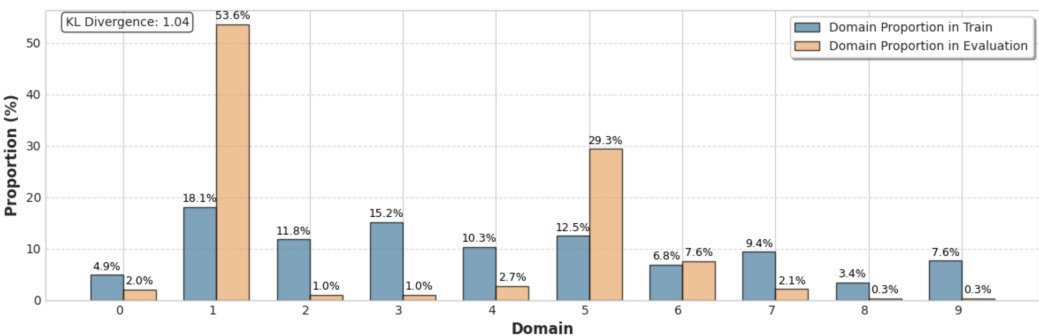

Figure 7: Comparison between domain proportions in training versus evaluation data (KL Divergence: 1.04). Our method strategically reweights domain distributions during training to optimize performance, notably increasing the representation of Domains 1 and 5 while reducing emphasis on Domains 2, 3, 4, 7, 8, and 9 compared to their evaluation proportions.

Table 8: Mapping between original categories and regrouped clusters

| Original Category | Regrouped Cluster |
| --- | --- |
| brainstorming | Cluster 0: General knowledge and open-ended questions covering a wide range of topics from science, technology, to basic concepts |
| classification | Cluster 1: Music-related queries focusing on instrument classification, musical theory, and instrument comparisons |
| closed QA | Cluster 2: Information extraction and summarization tasks about various topics including companies, historical figures, and specific domains |
| generation | Cluster 3: Classification tasks primarily involving animals, colors, household items, and biological categorizations |
| information extraction | Cluster 4: Sports-related queries spanning multiple disciplines including golf, F1 racing, Olympics, and team sports |
| open QA | Cluster 5: Entertainment and pop culture queries about movies, TV shows, musicians, artists, and historical personalities |
| summarization | Cluster 6: Lifestyle and creative brainstorming queries covering diverse topics from home improvements to personal recommendations |
| creative writing | |

evaluation proportion is higher at 53.6%, and Domain 5 stabilizes at approximately 0.200 despite its 29.3% evaluation proportion. This deliberate partial convergence suggests that optimal performance requires a strategic balance—influenced by but not identical to the evaluation distribution.

## H    CLUSTERING INTERPRETATION

In this section, we provide some interpretation about the groups discovered via clustering.

Table 8 shows the difference in groups before and after clustering on the Dolly-15k dataset. The left column displays the initial eight categories used to organize the text dataset during collection: brainstorming, classification, closed QA, generation, information extraction, open QA, summarization, and creative writing. The right column shows the seven distinct clusters that emerged when applying our clustering algorithm to the entire corpus. Interestingly, rather than following the original task-based boundaries, these clusters primarily organized around content domains, and subject matter, and subject length. This suggests that semantic content features may be more salient for learning features than the original task-based categorization framework, potentially offering new insights into how language models naturally organize information.

