# OpenReview forum: "R&B: Domain Regrouping and Data Mixture Balancing for Efficient Foundation Model Training"
_ICLR.cc/2026/Conference — Submitted to ICLR 2026_

### Official Review · Reviewer_mCFW · 2025-10-17

**Soundness:** 4
**Presentation:** 3
**Contribution:** 3
**Rating:** 8
**Confidence:** 3

**Summary:**

The paper proposed a novel gradient-based method (R&B) a two-stage framework for efficient data mixture optimization. The paper repartition as Regroup training data into semantically coherent clusters using embedding similarity, and it optimize domain weights as Balance to get individual domain contributions and cross-domain relationships, leveraging domain gradients computed via training. The paper validate the superiority of the proposed method across five diverse data settings.

**Strengths:**

- The paper is well-organized and easy to read for the main contributions. The theoretical justification are also thoroughly presented in the supplementary material.
- The proposed method makes use of more flexible and granular grouping method instead of the fixed and hand-crafted categorizations and it results in better performances.
- The proposed method achieves efficiency by avoiding extra computational costs because it uses gradient information derived during training.
- The proposed method is validated over multiple types of tasks and datasets from LLM to image classification.

**Weaknesses:**

- The regrouping and balancing framework yield new hyperparameters that must be tuned, which adds an extra hyperparameter optimization burden.
- According to Fig. 3, the performance can vary heavily depending on the number of clusters in certain datasets, indicating that this parameter has a highly sensitive impact on the overall performance.
- As the number of clusters (domains) increases, the computation and memory cost of constructing and using a Gram matrix may become burdensome. the scalability and generalization of this approach under multiple domains deserve further studies.
- For tasks with highly diverse domain distributions, semantic clustering might blur distinct domain characteristics, potentially causing overfitting or under-representation of certain domains.
- Is there any plan to release the official code?

**Questions:**

- In line 140, the capital delta is not defined.
- Why is the capital delta's dimension (m-1)?
- Is it possible to use mutual information instead of gram matrix?

---

> ### Author Response · Authors · 2025-11-22
>
> We thank reviewer mCFW's constructive feedback on our paper. Below, we address the specific questions raised in the review.
>
> # New hyperparameters
>
> We agree that R&B introduces a small number of additional hyperparameters (e.g. $\lambda$). In practice, however, we find that these hyperparameters are extremely stable: we use the same hyperparameter across all datasets without task-specific tuning. We use $\lambda=3$ as given in Table 7 in the Appendix.
>
> # Performance and number of clusters
>
> Figure 3 (bottom row) illustrates that performance varies with the intrinsic separability of the dataset (the Silhouette score), not due to instability of the method. When domains are genuinely distinct, using more clusters improves performance; when the domains overlap, performance is flatter. This effect therefore reflects dataset structure rather than a brittle sensitivity of R&B.
>
> # Gram Matrix scalability
>
> Although the Gram matrix is of size $m \times m$, its cost is far smaller than that of other data-mixing methods, which must operate over $D_t$ training tokens and/or $D_e$ evaluation tokens. Moreover, $m$ is inherently limited by the dataset: there can never be more domains than examples, and each example contains many tokens in its context, so $m$ is always orders of magnitude smaller than both $D_t$ and $D_e$. In practice, the number of domains $m$ is usually orders of magnitude smaller than the number of examples, and in all of our settings $m<100$, which is negligible compared to the compute cost of training. We provide a detailed compute analysis in Appendix D.
>
> # Tasks with highly diverse domains
> This is a valid concern, and R&B focuses on improving datasets that rely on human-defined domain labels or noisy data. Empirically, we observe that diverse domains (e.g. historical facts, geography and travel) are assigned to different domains, as shown in Figure 1. This shows that we only regroup examples whose corresponding embeddings are close, and keep semnatically distinct clusters separate.
>
> # Code Release
> Yes, we plan to release the full implementation, including the extractor, regrouping pipeline, and training code.
>
> # Additional Questions
>
> > In line 140, the capital delta is not defined.
>
> We take $\Delta^{m-1}$ to be the $(m-1)$-dimensional probability simplex. In other words
> $$\Delta^{m-1} = \{\alpha\in \mathbb{R}^m: \alpha_i \geq 0 , \sum_{i=1}^m \alpha_i = 1 \}$$. We will clarify this in the revision.
>
> > Why is the capital delta's dimension (m-1)?
>
> We take $m-1$ here because the simplex only has $m-1$ degrees of freedom. In this case, with $m$ being the number of domains, the sampling proportions must add up to 1, so the $m$-th domain's proportion is just 1 subtracted from all the others.
>
> > Is it possible to use mutual information instead of gram matrix?
>
> This is certainly an interesting idea. In principle, one could define pairwise mutual information over gradients by treating a gradient $g$ as a high-dimensional random variable and computing $I(B_{ij}, g)$, where $B_{ij}$ indicates whether a gradient came from domain $i$ or $j$. However, this may introduce additional complexity and place assumptions on the distribution of the random variable. In practice, we find that our Gram matrix formulation aligns well with our regret analysis (Section 3) and is straightforward to compute.

---

> > ### Comment · Reviewer_mCFW · 2025-11-28
> >
> > Thanks for your detail comments for solving my concerns. My major concerns are solved and I will keep my initial rating.

---

### Official Review · Reviewer_k3cV · 2025-10-27

**Soundness:** 2
**Presentation:** 2
**Contribution:** 3
**Rating:** 4
**Confidence:** 3

**Summary:**

The paper proposes a novel framework to address the limitations of existing data mixture optimization methods, which often fail to capture semantic relationships and suffer from computational scalability issues. Specifically, the proposed approach consists of two stages: (1) it partitions the training dataset into fine-grained domains based on semantic similarity, a step named to as __Regroup__; and (2) it leverages a Gram matrix of domain gradients to optimize the overall data composition, a step named __Balance__. The method is supported by theoretical analysis and demonstrates consistent effectiveness across five diverse benchmark datasets, spanning natural language understanding, reasoning, and multimodal tasks.

**Strengths:**

1. The proposed method is intuitive, straightforward, and effective. The authors provide a clear and well-structured explanation of the conceptual framework, making the approach easy to follow.

2. The visual illustrations in the paper are highly informative and enhance the reader’s understanding. For instance, **Figure 1** clearly defines the mixing strategy and highlights the benefits of optimizing both the proportions and the number of domains, while **Figure 2** vividly visualizes the key steps of the training process.

3. The proposed method is comprehensively validated across multiple benchmark tasks—including natural language understanding, reasoning, and multimodal datasets

**Weaknesses:**

1. The novelty of the proposed method is  ambiguous, and its distinction from prior work—both conceptually and technically—is not clearly articulated. The third paragraph (Lines 45–70) attempts to highlight the limitations of existing approaches as motivation but does not cite any relevant literature. For example, in **Line 48**, the statement __“Optimizing the proportions … the general predefined domains”__ lacks a citation, making it unclear whether this is an original claim or one established in prior studies. Similarly, in **Line 53**, the phrase __“Existing approaches …”__ is not supported by references. Beyond the introduction, in **Section 3.2 (Line 186)**, the statement __“Intuitively, data points … in both clusters”__ raises the same concern—whether the idea of gradient alignment is novel to this work or derived from earlier research. Overall, the paper presents several claims without sufficient clarification of what constitutes its genuine contribution versus what is grounded in prior literature.

2. The paper’s discussion of the **data mixture optimization problem** lacks comprehensiveness and is not reader-friendly for those unfamiliar with the field. For instance, the first paragraph (Lines 32–35) is intended to introduce the background but provides only an abstract definition without concrete examples or an explanation of the practical benefits of data mixing. The motivation would be clearer if the authors illustrated how data mixing influences model performance—with or without optimization—and its real-world impact. Furthermore, in **Line 43**, the term __“the optimal groups of data mixing”__ is vague. The paper does not specify the evaluation metric used to assess a data mixing strategy—whether it is training loss, test loss, or another measure such as accuracy. The rationale for using loss as the optimization criterion, instead of accuracy-based metrics, should also be clarified. Finally, the paper does not discuss whether the proposed mixing strategy addresses in-distribution versus out-of-distribution generalization challenges.

3. The claimed computational advantage of the proposed method—with only a 0.01% additional compute overhead—is not clearly explained in the Method section. In **Line 80**, the description is confusing: the phrases __“only requires an additional overhead”__ and __“cuts more than 99% FLOPs”__ appear contradictory. The paper should explicitly clarify which component constitutes the computational bottleneck, how each computation cost is estimated, and what the reported overhead quantitatively represents. Additionally, the relationship between the overhead and FLOPs reduction should be rigorously defined and empirically supported. Given that computational efficiency is a key claimed advantage of this work, the authors are encouraged to dedicate at least one subsection in the Method section to describe the computational analysis in detail, and another subsection in the Experiment section to explain the experimental setup and the procedure used for measuring and validating the computation cost.

**Questions:**

- Overall, could the authors provide a more explicit comparison—both conceptual and technical—between their framework and existing data mixture optimization or domain partitioning methods?

- Are the claims in Lines 48–53 (e.g., “Optimizing the proportions … predefined domains” and “Existing approaches …”) based on prior literature or newly proposed in this paper? Please provide explicit citations where applicable.

- In Section 3.2 (Line 186), is the idea of gradient alignment between clusters an original contribution, or is it adapted from prior research?

-  What evaluation metric is used to assess the quality of a data mixing strategy—training loss, test loss, or accuracy—and why was this metric chosen?

- Does the proposed mixing strategy explicitly address in-distribution and/or out-of-distribution generalization challenges? If so, how?

- How can the claim of “only an additional overhead” be reconciled with “cuts more than 99% FLOPs”? What exactly is being reduced or added computationally?

- Could the authors provide a breakdown or formula showing how computation cost is estimated for each module?

- Would the authors consider adding a dedicated subsection in the Method or Experiment section explaining how computational efficiency was measured and validated?

---

> ### Author Response · Authors · 2025-11-22
>
> We thank Reviewer k3cV's constructive feedback on our paper. Below, we address the specific questions raised in the review.
>
> # Novelty of our method
>
> > Overall, could the authors provide a more explicit comparison—both conceptual and technical—between their framework and existing data mixture optimization or domain partitioning methods?
>
> Our primary contribution is a **substantial reduction in compute overhead for data-mixing methods while maintaining competitive performance**. In Lines 52–53, we highlight that **as the number of domains grows, existing methods become computationally prohibitive** because they require either:
>
> * **Forward passes over a validation set per domain** (e.g., Skill-It, Aioli), or
> * **Domain-specific gradient computations against evaluation data** (e.g. DGA), which are often full gradients.
>
> R&B requires neither evaluation forward passes nor evaluation gradients. Instead, it uses **training gradients already computed during standard training**, enabling a _dual-purpose_ use: both parameter updates and domain-weight updates.
>
> We have provided a detailed computation-cost comparison in Appendix D.4–D.5, providing concrete FLOP breakdowns for each baseline.
>
> > Are the claims in Lines 48–53 (e.g., “Optimizing the proportions … predefined domains” and “Existing approaches …”) based on prior literature or newly proposed in this paper?
>
> We will revise the introduction to clarify this.
> The idea of optimizing proportions across semantically defined domains is established in prior work, including:
> * DGA  (Fan et al., 2024): domain reweighting using gradient alignment
> * CLIMB (Diao et al., 2025): mixture optimization via small proxy models
>
> These methods differ substantially from R&B:
> * DGA [1] computes gradient alignment against only a single evaluation batch, which requires that batch to be sufficiently large to represent the entire evaluation distribution. As a result, the evaluation batch must be large and diverse, adding substantial compute overhead and making the method sensitive to batch size.
> * CLIMB [2] requires training proxy models for mixture identification.
> * R&B avoids both costs through online, evaluation-free updates.
>
> > In Section 3.2 (Line 186), is the idea of gradient alignment between clusters an original contribution, or is it adapted from prior research?
>
> Gradient alignment has been explored in DGA [1]. However, as stated before,  DGA requires:
> * evaluation data gradients, and
> * full gradients per domain
>
> both of which increase overhead substantially. On the other hand, R&B's innovation is showing that using only last-layer gradients are sufficient, and do not require evaluation data gradients because we only use training data gradients (plus validation domain proportions) to perform reweighting. This is the main reason that R&B achieves significantly lower compute overhead than the other methods.
>
> > What evaluation metric is used to assess the quality of a data mixing strategy—training loss, test loss, or accuracy—and why was this metric chosen?
>
> We report (validation) loss as well as test accuracy for our metrics.
> * Validation loss (Table 1) is the objective directly targeted by data-mixture optimization, as it reflects model fit across multiple evaluation domains.
> * Accuracy (Table 3) is reported for downstream task generalization (e.g., CLIP tasks).
>
> We further add additional experiments showing that our method improves accuracy on downstream reasoning tasks. In the table below, we evaluate the performance of each data-mixing method on downstream tasks (GPQA-diamond, GSM8K, Math500), and their associated FLOP overhead over running stratified data mixing. All models are Qwen2.5-1.5B models trained on S1-1.1k reasoning traces, using ModernBERT (149m parameters) as our embedding model. Our formulas for calculating FLOP overhead are given in Appendix D:
>
> | Method (ModernBERT) | GPQA-Diamond | GSM8K | Math500 | **Average** | **Regroup Overhead (%)** | **Reweight Overhead (%)** | **Total % Overhead** |
> | ------------------- | ------------ | ----- | ------- | ----------- | ------------------------ | ------------------------- | -------------------- |
> | **aioli**           | 25.7         | 65.1  | 54.4    | 48.4        | 1.61                     | 29,459.00             | 29,460.61            |
> | **dga**             | 29.2         | 64.6  | 51.2    | 48.3        | 1.61                     | 3.50                  | 5.11                 |
> | **rnb**             | 27.8         | 63.6  | 53.0    | 48.1        | 1.61                     | **6×10⁻⁵**        | **1.61**                 |
> | **skillit**         | 25.7         | 65.1  | 51.6    | 47.5        | 1.61                     | 7,915.00              | 7,916.61             |
> | **stratified**      | 27.2         | 56.3  | 49.8    | 44.4        | 0                        | 0                         | 0                    |

---

> ### Author Response · Authors · 2025-11-22
>
> The above table shows that R&B is able to attain competitive performance with other methods, while requiring substantially less compute overhead than other data mixing methods.
>
> > Does the proposed mixing strategy explicitly address in-distribution and/or out-of-distribution generalization challenges? If so, how?
>
> Yes, the SupNatInst dataset is specifically for in-distribution task generalization, and SupNatInst-Test is specifically for out-of-distribution task generalization. The results are provided in Table 1.
>
> # Computational Cost Measurement
>
> > How can the claim of “only an additional overhead” be reconciled with “cuts more than 99% FLOPs”? What exactly is being reduced or added computationally?
>
> Thank you for pointing this out. We specifically state that R&B reduces computational overhead _over existing data-mixing approaches_.
> * R&B vs. Static uniform sampling: R&B adds a small overhead relative to static sampling (since static sampling costs zero).
> * R&B vs. existing data-mixing methods (DGA, Aioli, Skill-It): R&B reduces >99% of the FLOPs because it removes evaluation-set gradients and domain-level forward passes.
>
> As we demonstrate in Table 1, **we are orders of magnitude more efficient than existing approaches** because we do not require evaluation forward passes or gradients to perform reweighting. Thus, while we do have a small amount of overhead compared the simple stratified sampling method approach, we are still massively more efficient in FLOPs than existing approaches.
>
> > Could the authors provide a breakdown or formula showing how computation cost is estimated for each module
>
> We provide a breakdown of the computation costs in Appendix D.
>
> > ... consider adding a dedicated subsection in the Method or Experiment section explaining how computational efficiency was measured and validated?
>
> Yes, also see Appendix D.
>
> [1] Fan, S., Grangier, D., and Ablin, P. Dynamic Gradient Alignment for Online Data Mixing. arXiv 2410.02498 (2024).
>
> [2] Diao, S., Yang, Y., Fu, Y., Dong, X., Su, D., Kliegl, M., Chen, Z., Belcak, P., Suhara, Y., Yin, H., Patwary, M., Lin, Y. (C.), Kautz, J., and Molchanov, P. CLIMB: CLustering-based Iterative Data Mixture Bootstrapping for Language Model Pre-training. arXiv 2504.13161 (2025).

---

### Official Review · Reviewer_FWbm · 2025-10-27

**Soundness:** 2
**Presentation:** 3
**Contribution:** 1
**Rating:** 2
**Confidence:** 5

**Summary:**

The paper introduces R&B, a two-stage framework to optimize data mixtures. The first stage, Regroup, aims to deal with the limitation of suboptimal human categorizations based on skills . It is a one-time offline process that re-partitions the training data into fine-grained, semantically coherent clusters using k-means on data embeddings from an existing embed model. The second stage, Balance, is an online algorithm that dynamically adjusts the sampling proportions of these new domains during training. It avoids expensive re-evaluations by reusing existing last-layer training gradients, computing a Gram matrix to align the sampling distribution with a target evaluation distribution. The authors claim R&B matches or outperforms state-of-the-art data-mixing strategies with negligible computational overhead.

**Strengths:**

1. The paper notices a key problem in practice: the domain predefined by human may be suboptimal.
2. The paper empirically shows that design domain weights based on regrouped domains could improve performance.
3. The paper proposes an efficient way to adjust domain weights during training through accumulating gradients.
4. The experiments cover multiple tasks.

**Weaknesses:**

1. Main weakness is that this paper is too similar to DGA [Fan et al., 2024a], although the authors claim that DoGE [Fan et al., 2024b] is the most relevant one. Specifically, DGA (1) uses embeddings from an existing embed model to represent data for self-clustering; then (2) uses the inner product of domains' gradients to compute alignment across domains for computing domain weights. These two steps are almost the same as the two key steps in this paper: regroup and reweight. The main differences are (1) R&B uses the last layer's gradient while DGA uses the full gradient, and (2) R&B uses accumulated gradients during training while DGA computes the gradient on the extra validation set. I think these two differences are minor.
2. This paper doesn't fully explain the benefit of using the last layer's gradient compared with the full gradient or other layers' gradients. I mean except memory overhead, why R&B using the last layer's gradient show better performance than DGA with full gradient?
3. The paper doesn't explain well why accumulated gradients can show better performance than gradients computed on extra evaluation set compared with DGA.
4. The main weakness of accumulated gradient during training is that some domains with low weights may have few samples leading to noisy/inaccurate gradients. How to avoid such a situation?
5. In the experiments, the paper only shows validation loss but has no downstream task performance, especially for the language model, which is what people care about more and is also a standard comparison way in data mixing papers [Xie et al., 2023a, Fan et al., 2024b, Liu et al., 2024].

**Questions:**

1. For the data regrouped into finer-grained domains, can you really get their new domain names like 'Historical Text', 'Music Theory' as shown in Figure 1 (1)?
2. Compared with the embedding-based data mixing method [1], what's the advantages and disadvantages of using an existing embed model instead?


[1] Wanyun Xie, Francesco Tonin, and Volkan Cevher. Chameleon: A flexible data-mixing framework for language model pretraining and finetuning. ICML, 2025.

---

> ### Author Response · Authors · 2025-11-22
>
> We thank Reviewer FWbm's constructive feedback on our paper. Below, we address the specific questions raised in the review.
>
> # Similarity to DGA
>
> We emphasize that R&B achieves substantial cost savings during reweighting, which differs significantly from that of DGA. DGA computes a gradient for each training domain, and aligns each domain’s gradient with a **single intermediate gradient** computed on a small held-out validation set. This design introduces two key limitations:
> * The alignment signal is sensitive to the batch size and representativeness of that small validation set.
> * In many real-world scenarios, computing gradients on the target data is infeasible. For example, the evaluation dataset may be large (adding additional runtime) or privacy-restricted.
>
> R&B avoids both issues by relying solely on in-loop training gradients together with **aggregate validation proportions**, rather than explicit validation-set gradients. This makes our method applicable to broader settings and yields a more stable, multi-domain alignment signal than DGA’s single-batch alignment to $D_{spe}$.
>
> We include additional experimental results showing performance of DGA as a function of the batch size used versus R&B. We finetune Qwen2.5-0.5B base models on the NaturalInstructions dataset (we cluster with 30 domains) and evaluate on four downstream tasks (ARC-C, ARC-E, PIQA, Winogrande). We observe that DGA's performance improves with larger batch size; however, this improvement comes with significant computational cost (up to 15% with batch size 8). In contrast, R&B maintains competitive accuracy while incurring only 0.0014% overhead, which is several orders of magnitude cheaper than DGA's best-performing setting.
>
> | Method                | arc_challenge (%) | arc_easy (%) | piqa (%)  | winogrande (%) | Average (%) | **% Overhead** |
> | --------------------- | ----------------- | ------------ | --------- | -------------- | ----------- | -------------- |
> | **rnb**               | 31.14             | **67.21**    | 70.29     | 56.35          | 56.25       | **0.0014**     |
> | **dga (grad size 1)** | 30.38             | 65.99        | 70.35     | 57.30          | 56.00       | 1.93           |
> | **dga (grad size 2)** | **31.74**         | 66.50        | 70.57     | 57.06          | **56.47**   | 3.875          |
> | **dga (grad size 4)** | 30.89             | 66.25        | 70.78 | **57.54**      | 56.36       | 7.72           |
> | **dga (grad size 8)** | 31.23             | 66.67        | **70.95** | 56.59          | 56.36       | 15.44          |
> | **stratified**        | 31.06             | 65.45        | 70.46     | 56.35          | 55.83       | —              |
>
>
> # Last-Layer Gradient vs Full Gradient
>
> We argue that last-layer gradients provide a sufficiently strong signal for estimating how each domain influences model performance. More broadly, R&B offers a flexible **compute–performance tradeoff**: one can use deeper or full gradients at the cost of additional compute, or last-layer gradients for minimal overhead while still enjoying good performance. In our experiments, we chose to highlight the latter regime specifically to address the compute bottleneck of prior data mixing approaches.
>
> To validate this choice, we conducted an ablation comparing last-layer gradients versus full-gradients within R&B. We train Qwen2.5-1.5B models trained on S1-1.1k reasoning traces and evaluate on three downstream tasks (GPQA-diamond, GSM8K, Math500). We observe that using last-layer gradients in our method substantially outperforms using full gradients in our method.
>
> | Method            | GPQA-Diamond   | GSM8K  | Math500 | Average |
> | ----------------- | ------ | ------ | ------- | ------- |
> | **Full Gradient** | 0.2727 | 0.6399 | 0.4960  | 0.4695  |
> | **Last Layer**    | 0.2778 | 0.6368 | 0.5300  | **0.4815**  |
>
> # Why accumulated gradients outperform evaluation gradients.
>
> We attribute this to the fact that DGA computes gradients using a small, separate evaluation dataset, which is high-variance and sensitive to batch size. R&B accumulates gradients from each domain over an entire time window, which provides **stronger signal** of the true gradient of each domain.
>
> # Avoiding inaccurate gradients for domains with low weights
>
> We generally avoid this problem by ensuring that the amount of time between rounds (the time spent accumulating gradients) is large enough to provide sufficient signal about the domain's gradient direction.
>
> # Downstream task performance
>
> We provide additional experimental results for downstream tasks in both the instruction following setting (ARC-C/ARC-E/PIQA/Winogrande) and in the reasoning setting (GSM8K/MATH500/GPQA-Diamond), as above. We find that R&B is able to match performance with DGA while being substantially more compute efficient.

---

> > ### Author Response · Authors · 2025-11-22
> >
> > # Additional Questions
> > > For the data regrouped into finer-grained domains, can you really get their new domain names like 'Historical Text', 'Music Theory' as shown in Figure 1 (1)?
> >
> > We clarify: names are post-hoc heuristics (by inspecting the grouped clusters and observing the resulting data examples). They are only a means to interpret the clusters that arise from the Regroup stage.
> >
> > > Compared with the embedding-based data mixing method [1], what's the advantages and disadvantages of using an existing embed model instead?
> >
> > We thank the reviewer for providing this reference. We believe there are two distinctions to the embedding-based data mixing method of Chameleon [1]:
> > * The advantage of using an existing embedding model is to obtain **high quality embeddings** which can lead to better domain boundaries of the data. [1] proposes to use embeddings to estimate domain affinities, but importantly does not regroup data into new domains, which may still be suboptimal for data mixing. In section 3 of our paper discuss why **suboptimal grouping of data can lead to worse model performance**, supported with theoretical and empirical justifications.
> > * The model we used was ModernBERT, a 149m-parameter embedding model. It is slightly larger than the smallest embedding models used in [1], in practice it is relatively cheap compared to the overall cost of training a model.
> >     * For example, using formulas from Appendix D, we find that computing the embeddings for reasoning traces takes up only **1.6%** of the total compute for training the model on such traces.
> > * Moreover, we **can pay a one-time fixed cost** by performing the embeddings and reclustering the data to obtain new domains, which can be appropriately applied to any data-mixing method.

---

### Official Review · Reviewer_pCXM · 2025-10-29

**Soundness:** 2
**Presentation:** 2
**Contribution:** 1
**Rating:** 4
**Confidence:** 4

**Summary:**

The paper presents R&B (Regroup & Balance), a two-stage framework designed to optimize data mixtures efficiently during foundation model training. It tackles two main issues in existing approaches: their dependence on broad, human-defined data domains and the heavy computational cost of reweighting those domains. In the Regroup stage, R&B clusters training data into fine-grained, semantically coherent groups using embeddings. In the Balance stage, it dynamically adjusts data mixture proportions during training by using last-layer gradients that are already computed, eliminating the need for additional evaluation passes. A Gram matrix of domain gradient similarities guides a softmax-based update of sampling weights. Experiments across language, reasoning, and multimodal benchmarks show that R&B achieves equal or superior performance compared to state-of-the-art methods while cutting reweighting computation costs by over 99%.

**Strengths:**

1. Faster runtime
2. Well-motivated domain redefinition
3. Experimental setup

**Weaknesses:**

1. Dependence on embedding quality: the entire Regrouping stage depends on the quality of the text embeddings used for clustering. It remains unclear how sensitive the framework is to the choice of this model. If the embeddings fail to capture the salient semantic features for a given task, the resulting clusters may be suboptimal, leading to poor downstream performance.
2. The reasoning datasets should be evaluated on reasoning tasks, such as MMLU, GSM8K, MATH-500, etc., while the current evaluation uses validation loss.
3. The text can be clearer. For example, the caption of Algorithm 1 states "Data selection" but, as far as I understand, you are performing domain reweighing, not filtering any data.
4. The theoretical difference with DoGE and DGA is sometimes overstated, i.e., "The key innovation of R&B lies in its use of gradient information to dynamically adjust sampling priorities", which could also apply to DoGE/DGA.
5. Some references are not discussed, for instance [1] use domain embeddings and Gram matrix computation and [2] also performs clustering.

[1] Xie, W., Tonin, F., & Cevher, V. (2025). Chameleon: A Flexible Data-mixing Framework for Language Model Pretraining and Finetuning. ICML.
[2] Ling, Zhenqing, et al. "Diversity as a reward: Fine-tuning llms on a mixture of domain-undetermined data." arXiv preprint arXiv:2502.04380 (2025).

**Questions:**

1. What is the purpose of Lemma 3.1? The regrouping step to me seems quite straightforward (embed -> kmeans) and clear already.
2. See weaknesses

---

> ### Author Response · Authors · 2025-11-22
>
> We thank Reviewer pCXM's constructive feedback on our paper. Below, we address the specific questions raised in the review.
>
> # Empirical robustness across embedding models
>
> Our goal in Regroup is not to claim that any embedding model is equally suitable, but **rather that high-quality embeddings are inexpensive to obtain and sufficient for our algorithm to work well**. To that end, we specifically selected ModernBERT [1] as our data encoder because 1) it produces strong semantic embeddings, and 2) it has long-context support up to 8k tokens, which is essential for long CoT/reasoning traces.
>
> To illustrate this point, we provide an additional table that charts the performance of each method on downstream tasks (GPQA-diamond, GSM8K, Math500), and their associated FLOP overhead over running stratified data mixing. All models are Qwen2.5-1.5B models trained on S1-1.1k reasoning traces, using ModernBERT (149m parameters) as our embedding model. Our formulas for calculating FLOP overhead are given in Appendix D:
>
> | Method | GPQA-Diamond | GSM8K | Math500 | **Average** | **Regroup Overhead (%)** | **Reweight Overhead (%)** | **Total % Overhead** |
> | ------------------- | ------------ | ----- | ------- | ----------- | ------------------------ | ------------------------- | -------------------- |
> | **aioli**           | 25.7         | 65.1  | 54.4    | 48.4        | 1.61                     | 29,459.00             | 29,460.61            |
> | **dga**             | 29.2         | 64.6  | 51.2    | 48.3        | 1.61                     | 3.50                  | 5.11                 |
> | **rnb**             | 27.8         | 63.6  | 53.0    | 48.1        | 1.61                     | **6×10⁻⁵**        | **1.61**                 |
> | **skillit**         | 25.7         | 65.1  | 51.6    | 47.5        | 1.61                     | 7,915.00              | 7,916.61             |
> | **stratified**      | 27.2         | 56.3  | 49.8    | 44.4        | 0                        | 0                         | 0                    |
>
>
> This table shows that R&B is able to attain competitive performance with other methods, while requiring substantially less compute overhead than other data mixing methods.
>
> # Downstream Reasoning evaluations
>
> We provide results on downstream reasoning tasks in the above table.
> We focus on GPQA-Diamond, GSM8K, and Math500 because these reasoning-intensive benchmarks are sensitive to the training-set domain composition, making them the most relevant for evaluating data-mixing methods. We show that R&B achieves accuracy comparable to or better than more computationally intensive baselines, while maintaining the lowest total overhead among all adaptive data-mixing methods evaluated.
>
> # Clarity of Text
> We thank the reviewer for pointing out these ambiguities.
> 1) We agree that R&B performs reweighting rather than filtering. The phrase “Data selection” was intended to highlight that R&B adaptively selects sampling probabilities for each domain, not that it excludes data. We will  update the caption and description of Algorithm 1 to read “Domain reweighting via Regrouping and Balancing”.
> 2) We agree that R&B, DoGE, and DGA all leverage gradient information. A more accurate characterization of our contribution is that R&B **uses accumulated training gradients directly from the main model, rather than proxy or alignment gradients**, and shows that these gradients can serve a dual purpose: both for updating model parameters and for dynamically reweighting domains based on gradient statistics. Furthermore, we demonstrate that we are **massively cheaper in compute overhead for data mixing**. We will revise the corresponding sentence in the text.
>
> # Related Works
> We thank the reviewer for mentioning these related works. We will include a discussion of both references in the related work section:
> 1) Chameleon (Xie et al., ICML 2025) indeed computes a Gram (affinity) matrix over domain embeddings, but it still requires training a proxy model and does not regroup or reassign data points across domains.
> 2) Diversity as a Reward (Ling et al., 2025) focuses on the data-undetermined regime, where data have no known domain labels, and clusters samples to encourage diversity during fine-tuning. Our setting is complementary: R&B begins with predefined domain groups (as in large-scale pretraining corpora) and shows that regrouping them by embedding or gradient similarity yields more representative and balanced mixtures.

---

> > ### Author Response · Authors · 2025-11-22
> >
> > # On Lemma 3.1:
> > Lemma 3.1 is included to provide some theoretical justification as to why semantic regrouping should help. Many prior data-mixing or domain-weighting approaches implicitly rely on the intuition that “domains with similar gradients should be grouped together”. Our lemma formalizes this intuition by showing that the regret of choosing a sub-optimal clustering depends on inner products of gradients, which indicates clustering based on these gradients improves performance. This also aligns cleanly with the R&B algorithm, which weights the domain proportions based on gradient alignment.
> >
> >
> >
> > [1] Warner et al. Smarter, Better, Faster, Longer: A Modern Bidirectional Encoder for Fast, Memory Efficient, and Long Context Finetuning and Inference. arXiv 2412.13663 (2024).
> >
> > [2] Karen Spärck Jones. A Statistical Interpretation of Term Specificity and Its Application in Retrieval. Journal of Documentation (1972).
> >
> > [3] Sentence-Transformers. all-MiniLM-L6-v2 [Pre-trained model]. Hugging Face (accessed 2025). https://huggingface.co/sentence-transformers/all-MiniLM-L6-v2.

---

### Comment · Area_Chair_yf6V · 2025-11-26
**Reminder: Discussion Phase Engagement Needed**

Dear Reviewers:

As the deadline for the discussion phase is approaching in less than one week, could you kindly engage in the discussion with the other reviewers and provide your response to the authors’ rebuttal?

Best regards,

AC

---

### Author Response · Authors · 2025-12-03

# Common Response

We thank the AC and reviewers for their time and constructive feedback. The reviews raise four main themes: (i) novelty and relationship to prior work, (ii) the need for downstream reasoning evaluations, (iii) computational efficiency and scalability; and (iv) clarity, notation, and hyperparameters. We summarize our responses to each of the following:

## Novelty and relation to prior work
Reviewers pCXM, FWbm, k3cV asked how R&B differs from recent data-mixing methods that also use embeddings and gradients.
* R&B explicitly **regroups examples** into finer-grained domains **using inexpensive embeddings** before reweighting. Methods such as Chameleon [1] still operate on fixed domains and do not reassign examples across domains.
* Methods such as DGA [2] compute full gradients per domain against evaluation batches, which is both compute-heavy and sensitive to the choice and batch size of that evaluation set. In contrast, R&B **only requires last-layer training gradients and aggregate validation proportions** to estimate the optimal validation gradient. Furthermore, we demonstrate that **training gradients are dual-use: both for parameter updates and for domain weighting.**

## Downstream reasoning and instruction-following evaluations
Reviewers pCXM, FWbm, and k3cV requested downstream evaluations:
* For reasoning, we finetune Qwen2.5-1.5B and evaluate on GPQA-Diamond, GSM8K, and Math500. We show that R&B achieves competitive accuracy results with the best baselines, while its total compute overhead (regroup + reweighting) is **only 1.6%** compared to stratified sampling. Importantly, R&B's **reweight stage alone adds only 6×10⁻⁵% additional compute, which is orders of magnitude more efficient than other baselines.**
* For instruction-following, we finetune Qwen2.5-0.5B on NaturalInstructions and report accuracy on ARC-C/E, PIQA, Winogrande. Similarly, we show that R&B matches or slightly trails other methods in accuracy, while costing only **0.0014% additional overhead** and is also orders of magnitude cheaper.

## Compute overhead clarity and calculation
Reviewers k3cV and mCFW raised questions about our overhead claims and the role of the number of clusters/domains.

We will rewrite the relevant parts of the introduction and methods to distinguish:

* Overhead vs. static stratified sampling:
R&B inevitably incurs some overhead over a “do nothing” baseline; we now state this explicitly (e.g., total overhead ≈1.6% for S1-1.1k with ModernBERT).
* Overhead vs. other data-mixing methods: Relative to DGA, Skill-It, and Aioli, R&B **reduces compute in the reweighting stage by two to four orders of magnitude**, because it avoids repeated evaluation forward passes and evaluation gradients.

* We also clarify that although the Gram matrix is $O(m^2)$ in the number of domains, in practice we have $m < 100$, which is orders of magnitude smaller than both the number of model parameters and number of training/evaluation tokens. By contrast, other data-mixing methods scale with model size and/or require full evaluation passes. As a result, the Gram matrix computation is negligible compared to the overall training cost.
* Regarding the number of clusters, Fig. 3 shows that performance varies predictably with intrinsic dataset separability, allowing us to estimate an appropriate cluster count before training. This keeps cluster selection lightweight and does not require extensive hyperparameter tuning.

## Embedding dependence and last-layer vs. full gradient
pCXM and FWbm asked how sensitive Regroup is to embedding choice and the specific choice of using last-layer gradients.
* We clarify that the goal is not to show that any embedding model works, but that **high-quality, long-context encoders are inexpensive enough to be computed once, and reused many times**. We justify our choice of ModernBERT (149M) and note that the one-time embedding pass contributes only **≈1.6%** of total FLOPs for our reasoning setup.
* We find that using last-layer training gradients for reweighting is just *as good as full training gradients*, if not better. More broadly, R&B achieves **flexible compute-performance tradeoff**: practitioners can choose how many layers to include in the gradient signal based on their compute budget, while still benefiting from our dual-use gradient formulation.

## Clarity, notation, and hyperparameters
Finally, we will improve clarity and notation as requested:
* Algorithm 1 performs domain reweighting; we will adjust terminology accordingly.
* Explicitly defining $\Delta^{m-1}$ probability simplex
* Key hyperparameters (e.g., $\lambda$) are fixed across all datasets without per-task tuning.

[1] Wanyun Xie, Francesco Tonin, and Volkan Cevher. Chameleon: A flexible data-mixing framework for language model pretraining and finetuning. ICML, 2025.

[2] Fan, S., Grangier, D., and Ablin, P. Dynamic Gradient Alignment for Online Data Mixing. arXiv 2410.02498 (2024).

---

### Meta-Review · Area_Chair_ZCQQ · 2025-12-17

**Summary:**

Reviewer pCXM expressed concerns on dependence on embedding quality and missing evaluation on popular reasoning benchmarks. Reviewer FWbm pointed out lack of novelty compared to DGA and the missing explanation about the advantages of the last layer's gradient. Reviewer k3cV complained the clearness of novelty and motivation. Reviewer mCFW argued that this paper provides a more flexible and granular grouping method without human effort and supports the acceptance of this paper.

**Reviewer Concerns:**

The evaluation results on popular reasoning benchmarks raised by Reviewer pCXM are not supplemented in the revised paper. The novelty issue proposed by Reviewer k3cV are also not well addressed.

**Reviewer Scores:**

Reviewer pCXM: 4;

Reviewer FWbm: 2;

Reviewer k3cV: 4;

Reviewer mCFW: 8

---

### Decision · Program_Chairs · 2026-01-26

Reject